# A lignin-derived material improves plant nutrient bioavailability and growth through its metal chelating capacity

Qiang Liu [1,2], Tsubasa Kawai[2], Yoshiaki Inukai [3], Dan Aoki [2], Zhihang Feng [1], Yihui Xiao [1], Kazuhiko Fukushima[2], Xianyong Lin[1], Weiming Shi[4,5], Wolfgang Busch [6], Yasuyuki Matsushita [2,7] ✉ & Baohai Li [1] ✉

The lignocellulosic biorefinery industry can be an important contributor to achieving global carbon net zero goals. However, low valorization of the waste lignin severely limits the sustainability of biorefineries. Using a hydrothermal reaction, we have converted sulfuric acid lignin (SAL) into a water-soluble hydrothermal SAL (HSAL). Here, we show the improvement of HSAL on plant nutrient bioavailability and growth through its metal chelating capacity. We characterize HSAL's high ratio of phenolic hydroxyl groups to methoxy groups and its capacity to chelate metal ions. Application of HSAL significantly promotes root length and plant growth of both monocot and dicot plant species due to improving nutrient bioavailability. The HSAL-mediated increase in iron bioavailability is comparable to the well-known metal chelator ethylenediaminetetraacetic acid. Therefore, HSAL promises to be a sustainable nutrient chelator to provide an attractive avenue for sustainable utilization of the waste lignin from the biorefinery industry.

The biomass refinery sector represents an important component of a sustainable bioeconomy and has developed rapidly[1]. Lignocellulosic biomass is considered an ideal biomass for refining raw material for biofuels and other products, as the annual yield of about 200 billion tons doesn't pose competition for human food or animal feed[2]. Lignin accounts for 15–40% of the total carbon that is contained in lignocellulose biomass[3]. Lignin also constitutes the largest source of natural aromatic polymers and harbors great potential as a starting material for many biobased products[4]. However, due to its heterogeneous chemical structure with complex and variable linkages, lignin is usually regarded as an unfavorable interference factor and discharged as a waste product from the biorefining system[5]. Moreover, most of the lignin is not isolated but is rather burned on-site, which yields the most significant carbon emission from lignocellulosic

biomass refining[6]. To overcome this inefficiency in the biomass refining industry, a variety of technologies, including pyrolysis, base-catalyzed or acid-catalyzed hydrogenolysis, and oxidation have been developed for lignin valorization. The current lignin valorization products include low-cost carbon fibers, plant-derived plastics, fungible fuels, and commodity chemicals[6]. However, these lignin-derived materials only account for about 2% of the total industrial lignin (50 million tons), and hardly will be able to deal with the rapidly increasing volume of industrial lignin[7]. Without a sustainable route for utilization of this industrial lignin, the biomass refining industry will remain a considerable source of $CO_2$ emissions and waste while missing out on valorizing a substantial amount of lignin[8].

Hidden hunger or micronutrient malnutrition affects about one third of the world's population. This is mainly a result of insufficient

[1]MOE Key Laboratory of Environment Remediation and Ecological Health, College of Environmental and Resource Science, Zhejiang University, Hangzhou, Zhejiang, China. [2]Graduate School of Bioagricultural Sciences, Nagoya University, Nagoya, Japan. [3]International Center for Research and Education in Agriculture, Nagoya University, Nagoya, Japan. [4]International Research Centre for Environmental Membrane Biology, Department of Horticulture, Foshan University, Foshan, China. [5]State Key Laboratory of Soil and Sustainable Agriculture, Institute of Soil Science, Chinese Academy of Sciences, Nanjing, China. [6]Plant Molecular and Cellular Biology Laboratory, Salk Institute for Biological Studies, La Jolla, CA, USA. [7]Institute of Agriculture, Tokyo University of Agriculture and Technology, Fuchu-shi, Tokyo, Japan. ✉e-mail: fx7789@go.tuat.ac.jp; bhli@zju.edu.cn

micronutrients such as iron, calcium, and zinc in the calorie-rich staple crops that these people primarily consume[9]. A pivotal approach to reduce hidden hunger is to increase micronutrients in these staple crops through agricultural technologies or biofortification[10]. Iron deficiency represents one of the most causes of hidden hunger[11]. This is due to the low bioavailability of iron in alkaline soils, which account for 25–40% of surface arable land, as low iron bioavailability in soils not only severely reduces crop yield but also limits the potential increase of iron accumulation in food crops[12]. Micronutrient deficiency in food crops is becoming more severe because of excessive input of macro-nutrients such as nitrogen and phosphate and rising atmospheric $CO_2$ levels[13]. To counteract this, chemical compounds such as ethylene-diaminetetraacetic acid (EDTA) can be used as fertilizer additives to effectively improve metal ion bioavailability and elevate metal nutrient accumulation in food crops[14]. However, these fertilizer additives are costly and can cause significant environmental damage as they are non-biodegradable and thereby result in heavy metal pollution[15]. Lignin has several active functional groups, including aliphatic hydroxyl, carbonyl, and phenolic hydroxyl groups, as well as an unshared electron pair on the oxygen atom, all of which are implicated in metal ion chelation. Natural lignin-derived compounds are the largest supply source of the humic substances[16]. Humic substances in soils are natural chelates that increase metal nutrient bioavailability and contribute to avoiding metal nutrient deficiency phenotypes of plants[17]. Lignosulfonate has also been used to synthesize lignosulfonate-iron[18]. Taken together, this indicates that lignin-derived materials have the potential to be used as metal chelators and improve nutrient bioavailability. However, addition of industrial lignin to fertilizer did not draw much attention in the field of lignin valorization[19].

Sulfuric acid hydrolysis and enzymatic hydrolysis are two major strategies to produce monosaccharides in commercial biorefinery[20]. Sulfuric acid lignin (SAL, generated from sulfuric acid hydrolysis) and enzymatic hydrolysis lignin (generated from enzyme-mediated hydrolysis), are the major lignin byproducts from biorefinery[21]. Both types of lignin show very low solubility in both water and organic solvents due to their high molecular weights and highly condensed structures[22] (this is stronger in SAL due to the repolymerization and condensation reactions). This limits the valorization of the waste lignin and restricts the sustainable development of lignocellulosic biorefinery[23,24]. Therefore, SAL represents an acid-insoluble lignin residual portion obtained from polysaccharide conversion of lignocellulosic biomass by conducting acidolysis using concentrated sulfuric acid[25]. In a previous study, we converted high purity SAL that prepared by the Klason method to simulate industrial SAL, into water-soluble lignin fragments called hydrothermal sulfuric acid lignin (HSAL) by using an alkaline hydrothermal reaction[26]. Here, we showed that this hydrothermal reaction not only depolymerized SAL, but also strongly reduced methoxy groups and increased the portion of phenolic hydroxyl groups of lignin, conferring significant metal ion chelating capacity to HASL. We found that the addition of HSAL to low nutrient growth media and soils could significantly promote the growth of both monocot and dicot plant species including rice, corn, and *Arabidopsis thaliana*. Combining transcriptomics, genetics, and physiological approaches, we found that the promotion of HSAL on the root growth is primarily driven by enhancing metal nutrient bioavailability. The HSAL-mediated increase of iron bioavailability is comparable to the well-known but not-biodegradable metal chelator EDTA. Therefore, HSAL can serve as a promising sustainable metal chelator to replace synthetic chelating agents, and promises not only to promote the profitability and sustainability of biorefineries but also to sequester more carbon into the underground in future.

## Results

### Synthesis and characterization of water-soluble lignin fragments HSAL

We converted SAL into a liquid-form mixture by using an alkaline hydrothermal reaction at 280 °C for 2 h with an equal amount of NaOH. After removing inorganic reagents and filtering with a cellulose tube at a 3500 molecular weight cutoff, the liquid form of the hydrothermally treated SAL (HSAL) was collected and lyophilized to a powder[26]. As expected, the HSAL powder can be completely dissolved at room temperature in neutral water (The solubility is about 3 grams per 100 ml) (Fig. 1a).

Water solubility of lignin depends on the degree of depolymerization, de-methoxy, and phenolic hydroxyl content and carboxyl groups[27]. Since the carboxyl groups undergo an intense and thorough decarboxylation reaction under this hydrothermal reaction[28], the carboxyl groups can be ignored in HSAL[22]. In previous studies, we have shown that this hydrothermal reaction can cleave carbon-oxygen-carbon bonds, carbon-carbon bonds and alkyl-aryl ether bonds of the aromatic nucleus structure by using oligomeric lignin model compounds with $\beta$-O-4 linkage[22] and polymeric lignin model compounds[29]. This finding was partly supported by the reduction of the average molecular weight from 57KDa of SAL to 7.5 KDa of HSAL. This result suggested that the hydrothermal reaction effectively shears the carbon-oxygen-carbon bonds and carbon-carbon bonds connecting the lignin monomers to increase the depolymerization of SAL. To understand the overall effect of the hydrothermal reaction on the functional groups of SAL, Fourier Transform Infrared Spectroscopy (FT-IR) was employed to compare the functional groups between SAL and HSAL. FT-IR spectra showed that compared to SAL, absorption of methoxy groups at 1465 cm$^{-1}$ was decreased in HSAL, while absorption of hydroxy groups at 3200–3500 cm$^{-1}$ was increased sharply in HSAL (Fig. 1b). Quantification analysis showed that the methoxy group content of HSAL was 1.91 mmol/g, which was much less than 5.91 mmol/g in SAL (Fig. 1c). Quantification of hydroxy groups showed that the total phenolic hydroxyls amounted to 2.20 mmol/g in SAL and increased to 5.42 mmol/g in HSAL, while the amount of alcoholic hydroxy groups was greatly reduced in HSAL compared to SAL (Fig. 1d–f and Supplementary Fig. 1). Consistently, the proportion of methoxy groups per hundred phenylpropane units was 0.98–1.08 in SAL and was reduced to 0.32–0.35 in HSAL, while the proportion of phenolic hydroxyl groups per hundred phenylpropane units of HSAL (0.9–0.99) was much higher than that of SAL (0.36–0.4) and natural softwood lignin (0.15–0.30) (Fig. 1g). Taken together, these results supported the notion that the hydrothermal reaction not only breaks the carbon-oxygen-carbon bonds and carbon-carbon bonds between lignin monomers but also most likely cleaves the alkyl-aryl ether bonds of the aromatic nucleus structure to reduce the methoxy group and to vacate the positions for phenolic hydroxyl groups, which leads to the water solubility of HSAL.

### HSAL shows metal ion chelating capacity

Increased ratio of phenolic hydroxyl groups to methoxy groups of HSAL monomer units resembles the structure of humic substances that can act as natural nutrient chelators[30]. This prompted us to test the capacity of HSAL for metal chelation. We prepared different chelating complexes for HSAL with $FeSO_4$, $FeCl_3$, and $CaCl_2$ respectively. Unlike the porous loose structure of HSAL, the chelated complex of HSAL with $FeSO_4$ showed a fine sandy texture, while the chelated complexes of HSAL with $FeCl_3$ and $CaCl_2$ showed a fluffier look (Fig. 2a). The morphological change of these chelated complexes suggested the successful chelation of HSAL with metal compounds. We also performed elemental analysis of HSAL and HSAL-metal complexes. The results showed that iron content is clearly higher in HSAL-$FeSO_4$ and HSAL-$FeCl_3$, while calcium content is also much higher in

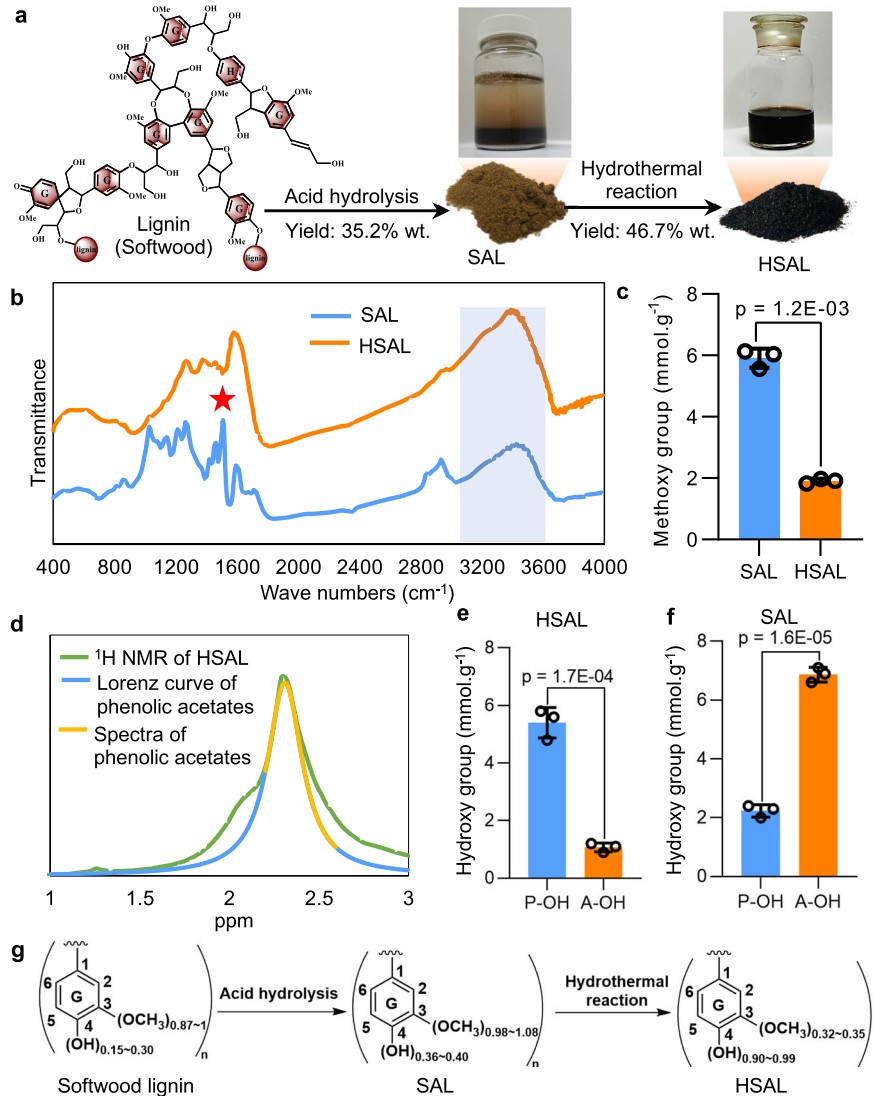

**Fig. 1 | Structural characteristics of hydrothermal sulfuric acid lignin (HSAL).**
**a** Synthesis of HSAL. During acid hydrolysis of lignocellulosic biomass, sulfuric acid lignin (SAL, yield was 35.2 wt% based on lignocellulosic biomass) was converted into a water-soluble lignin material HSAL (yield was 46.7 wt% based on SAL) with a mild hydrothermal reaction. **b**, Fourier transform infrared spectroscopy (FT-IR) spectra showed the difference of methoxy groups and hydroxy groups between SAL and HSAL. The red star indicates absorption of methoxy groups at 1465 cm⁻¹; the blue area indicates absorption of hydroxy groups between 3200 and 3500 cm⁻¹. **c** Methoxy group content in SAL and HSAL. **d** Measurement of phenolic hydroxy groups and alcoholic hydroxy groups of HSAL using nuclear magnetic resonance (NMR) spectra. **e** Quantification of hydroxy group content of HSAL in **d**. **f** Quantification of hydroxy group content of SAL. P-OH, phenolic hydroxyl group; A-OH, alcoholic hydroxy group. **g** Model diagram of lignin monomer structural units during HSAL synthesis with the change of methoxy groups and phenolic hydroxy groups. The graphs depict mean and standard deviation (error bars) and individual data points. Statistically significant differences between the means were analyzed with Student's *t*-test (*n* = 3 independent experiments) in (**c**, **e**, **f**).

HSAL-CaCl₂ compared to HSAL (Supplementary Table 1). This result supported that HSAL can chelate iron and calcium.

We further investigated the element composition and element valence state of HSAL and the chelated complexes of HSAL with different metal compounds using X-ray photo electron spectrometer (XPS) based on the specific binding energy. The binding energy for Fe(2p₃/₂) from FeSO₄ is between 711.9 and 712.3 eV, while that for Fe(2p₃/₂) from FeCl₃ is around 711.0–711.5 eV[31]. As shown in Fig. 2b, the Fe 2p peak for the chelated complexes of HSAL with FeSO₄ and FeCl₃ shifted to 709.7–710.6 eV and 723.3–723.9 eV, which were corresponding to Fe(2p₃/₂) and Fe(2p₁/₂) respectively, while no corresponding peak for HSAL was detected. These results indicate that FeCl₃ and FeSO₄ were successfully chelated by HSAL. The binding energy of Ca(2p₃/₂) of CaCl₂ is between 347.9 and 348.6 eV[32]. The binding energy of Ca 2p for the chelated complex of HSAL with CaCl₂ was resolved into two main peaks at 346.5–347.0 eV and 349.5–350.0 eV, which were

corresponding to Ca(2p₃/₂) and Ca (2p₁/₂) respectively (Fig. 2c). This result indicated that HSAL can also successfully chelate with CaCl₂.

To define the groups of HSAL involved in the chelation with metals, we performed FT-IR spectra analysis for HSAL and its chelated complexes with FeCl₃, FeSO₄, and CaCl₂ (Fig. 2d, e). The O-H stretch (H-bonded) of HSAL was at 3360 cm⁻¹, while it was shifted to 3320 cm⁻¹ and 3410 cm⁻¹ for the chelated complexes of HSAL with FeSO₄ and FeCl₃ (Fig. 2d). Moreover, two new band for each chelated complex of HSAL with FeSO₄ and FeCl₃ were observed at 665 cm⁻¹ and 1220 cm⁻¹, which correspond to Fe-O stretching vibrations and C–O bond of the guaiacol stretching respectively. Since C–O bond of the guaiacol structure primarily stemmed from phenolic hydroxyl groups in HSAL, phenolic hydroxyl groups likely served as the main chelating sites for HSAL binding with ferric iron. In the chelated complex of HSAL with CaCl₂, the band of the O-H stretching (H-bonded) vibration was

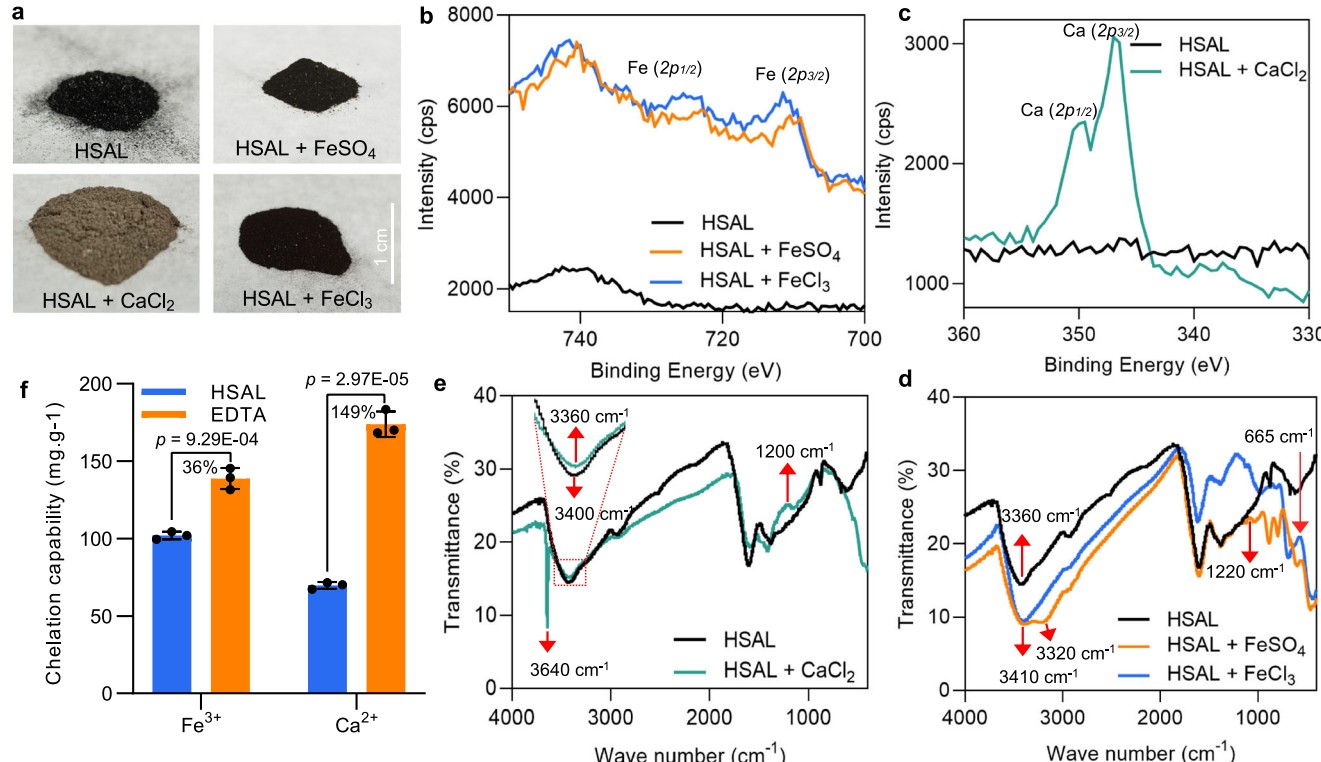

**Fig. 2 | HSAL shows a metal chelating capability. a** Representative images of HSAL powder and HSAL chelated with $FeSO_4$, $FeCl_3$ or $CaCl_2$ respectively. **b** X-ray photo electron spectrometer (XPS) spectra of HSAL and HSAL chelated with $FeSO_4$ or $FeCl_3$. The positions of Fe($2p_{1/2}$) and Fe($2p_{3/2}$) peak were located 723.3–723.9 eV and 709.7–710.6 eV respectively. **c** XPS spectra of HSAL and HSAL chelated with $CaCl_2$. The positions of Ca($2p_{1/2}$) and Ca($2p_{3/2}$) peak were 349.5–350.0 eV and 346.5–347.0 eV respectively. **d** FT-IR spectra comparisons of HSAL, HSAL chelated with $FeSO_4$ or $FeCl_3$. The FT-IR spectra at 3360 cm$^{-1}$ for the O–H band of HSAL was shifted to 3320 cm$^{-1}$ and 3410 cm$^{-1}$ for the chelated complexes of HSAL with $FeSO_4$ and $FeCl_3$ respectively. Two prominent bands at 665 cm$^{-1}$ and 1220 cm$^{-1}$ for the chelated complexes of HSAL with $FeSO_4$ and $FeCl_3$ were corresponding to Fe–O stretching vibrations and C–O of the guaiacol stretching respectively. **e** FT-IR spectra comparisons of HSAL and HSAL chelated with $CaCl_2$. The FT-IR spectra of the O–H band at 3360 cm$^{-1}$ for HSAL was shifted to 3400 cm$^{-1}$ for the chelated complexes of HSAL with $CaCl_2$. Two new bands at 1200 cm$^{-1}$ and 3640 cm$^{-1}$ were likely assigned to C–O of the guaiacol stretching and the calcium hydroxide stretching respectively. **f** Comparisons of metal chelating capability of HSAL and EDTA using complexometric titration analysis. The graphs depict mean and standard deviation (error bars). Statistically significant differences between the means were analyzed with Student's t-test ($n = 3$ independent experiments).

shifted from 3360 cm$^{-1}$ to 3400 cm$^{-1}$ (Fig. 2e). Moreover, in the chelated complex of HSAL with $CaCl_2$, two new bonds in the FT-IR spectrum were observed around 1200 cm$^{-1}$ and 3640 cm$^{-1}$, which can be likely assigned to the C–O of the guaiacol stretching and the O–H bonds (free hydroxyl) from calcium hydroxide[33] due to calcium chloride combining remaining hydroxide in the chelated complexes, respectively. These results demonstrated that the O–H bond is important for HSAL to chelate with $FeSO_4$, $FeCl_3$, and $CaCl_2$, while the C–O bond provided by phenolic hydroxyl group seems more specific for HSAL chelation with $FeCl_3$ and $CaCl_2$. To quantify the chelating capacity of HSAL, we measured the capacity of HSAL chelated with $FeCl_3$ and $CaCl_2$ using chelatometric titration analysis, and compared it to the well-known chelator EDTA. The results indicated that the capacity for HSAL to chelate $Fe^{3+}$ was 101.9 mg. g$^{-1}$, slightly lower than that of EDTA (138.8 mg. g$^{-1}$) ($p = 9.3E$-04). The capacity for HSAL to chelate $Ca^{2+}$ was 69.7 mg g$^{-1}$, while that of EDTA was 173.8 mg g$^{-1}$ (Fig. 2f) ($p = 3.0E$-05). As the average molecular weight of HSAL (7.5 kDa) is 25.6-fold higher than EDTA, equimolar amounts of HSAL can chelate more metals than EDTA. Taken together, these results demonstrated that HSAL has metal chelating capacity.

Since the inorganic phases might form crystalline hybrid materials due to the presence of free NaOH[34,35], we took HSAL and its chelated complexes with $FeCl_3$ as examples to perform X-ray powder diffraction analysis (XRPD) and Transmission Electron Microscope-energy dispersive X-ray spectrometer (TEM-EDS) analysis to examine whether the HSAL-metal chelating complexes could form hybrid materials composed by HSAL and inorganic phases. As shown in Supplementary Fig. 2a, the XRPD trace of HSAL and HSAL-$FeCl_3$ showed a smooth crest, which indicating the amorphous nature of HSAL and HSAL-$FeCl_3$. The TEM-EDS images of HSAL and HSAL-$FeCl_3$ complexes showed no bright spot aggregation distribution, which also supported the amorphous nature of HSAL and HSAL-$FeCl_3$ complexes (Supplementary Fig. 2b, c). Taken together, these results exclude the possibility that these chelated metal compounds form oxide crystals embedded in the polymeric matrix of HSAL.

## HSAL promotes root growth in multiple plant species

Metal chelators including synthetic EDTA and natural humic substances have been associated with enhanced plant growth under nutrient limited conditions[36]. To reveal a potential positive effect of HSAL on plant growth, we tested the effect of different concentrations of HSAL (control, 0.01%, 0.05%, and 0.1%) on the early growth of rice seedlings (*Oryza sativa* L. *ssp. japonica* cv. *Nipponbare*) grown in water without additional nutrients. All treatments with HSAL for 14 days after germination led to a substantial increase in root and shoot biomass, and root and shoot length in rice seedlings (Fig. 3a–e). The most pronounced increases (a 2.39-fold increase of root length ($p < 0.05$); a 1.30-fold increase of root biomass ($p < 0.05$) were observed with a concentration of 0.05% HSAL (Fig. 3b, c). A similarly increased pattern

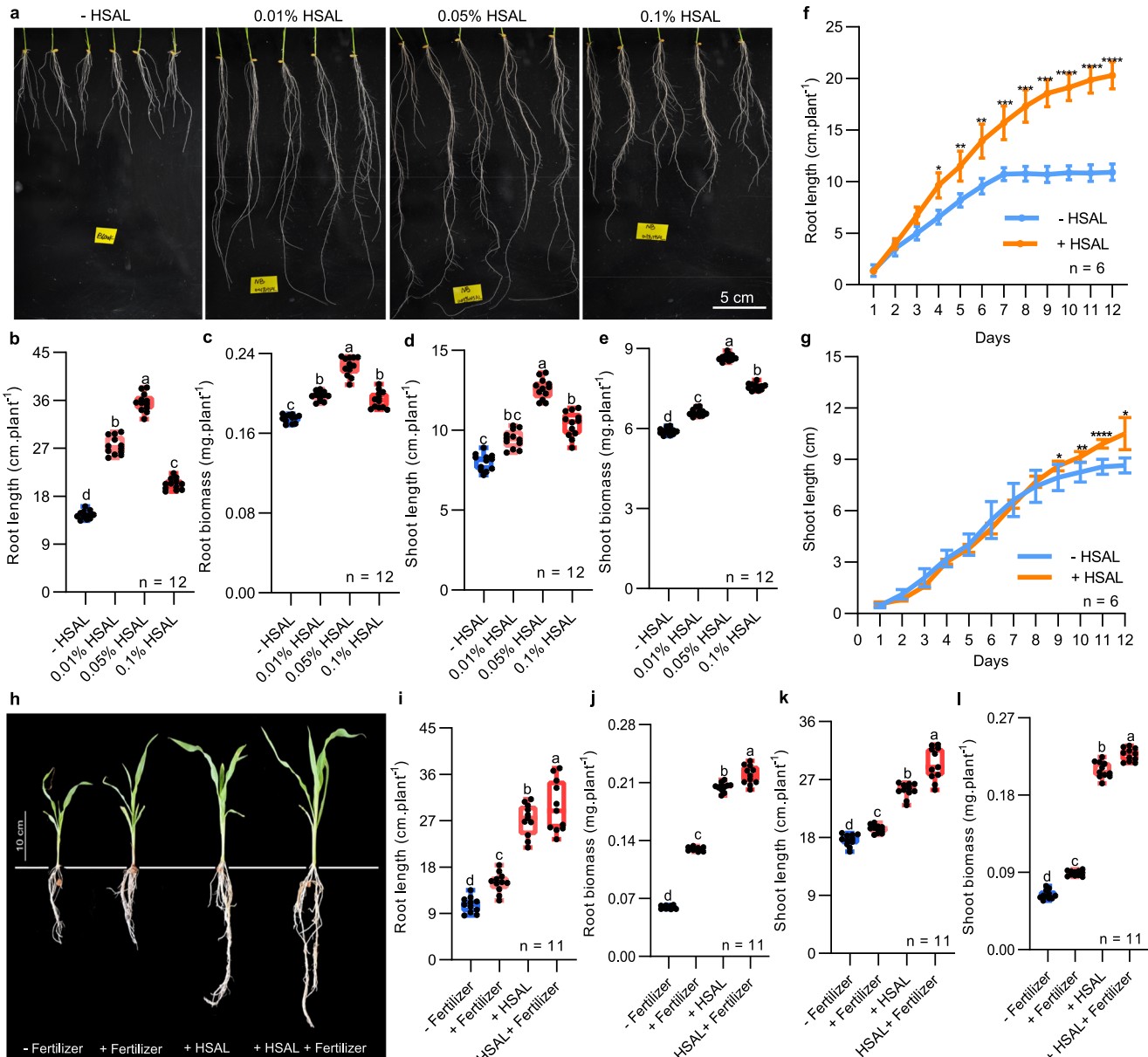

**Fig. 3 | HSAL additive significantly promotes root elongation in different plant species. a** Representative images showing rice roots (*Oryza sativa Nipponbare*) from seedlings treated with different concentrations of HSAL for 14 days. **b**, **c** Quantification of primary root length **b** and total root biomass **c** of seedlings in **a** using boxplots and individual data points. **d**, **e** Quantification of shoot length **d** and total shoot biomass **e** of seedlings in **a** using box plots and individual data points. **f**, **g** Growth curve on the root **f** and shoot **g** of *Nipponbare* between control and 0.05% HSAL treatments. Line graphs depict mean and standard deviation (error bars). Statistically significant differences between the means were analyzed with Student's *t*-test (*$p < 0.05$; **$p < 0.01$; ***$p < 0.001$; ****$p < 0.0001$) in **f**, **g**.

**h** Representative images of corn seedlings grown in soil treated with control solution (- fertilizer), 0.4% fertilizer (HYPONex), 0.4% HSAL, and 0.4% HSAL + 0.4% fertilizer, Scale bars = 10 cm. **i**, **j** Quantification of primary root length **i** and total root biomass **j** in **h**. **k**, **l** Quantification of shoot length **k** and total shoot biomass **l** in **h**. Boxes represent the 25th–75th percentile range, white lines represent medians and whiskers show the minimum–maximum range in **b–e**, **i–l**. Different letters indicate statistically significant differences based on one-way ANOVA and Tukey's *HSD* test analysis ($p < 0.05$). The number (n) of seedlings shown in the figure in **b–g**, **i–l**).

of HSAL was observed for the shoot (Fig. 3d, e). By measuring root length of *Nipponbare* rice grown under control conditions or with 0.05% HSAL each day, we found that differences in root length between the control and 0.05% HSAL could be observed starting 3 days after germination. Moreover, while roots stopped growing after 7 days in the control condition, they continued to grow at 0.05% HSAL (Fig. 3f). The promotion of shoot growth by 0.05% HSAL was detectable at a later point than that of the root (Fig. 3g). A similar pattern of HSAL growth promotion was observed in two other ecotypes of rice that we tested (*Oryza sativa L. japonica*. var. *Taichung 65* and *Kasalath*)

(Supplementary Figs. 3 and 4). These results indicated that HSAL mediated promotion of plant growth is common in different rice varieties.

To test the effect of HSAL on the growth of other plant species, we compared the growth of corn (*Zea mays* L.) seedlings in the soil comparing fertilizer and HSAL treatments (Fig. 3h). Compared to the seedlings grown in the control soil, root length was increased by 1.41-fold ($p < 0.05$), 2.56-fold ($p < 0.05$), 2.80-fold ($p < 0.05$) by 0.4% fertilizer, 0.4% HSAL and the combination of the treatments respectively (Fig. 3i). HSAL mediated growth promotion of corn seedlings were also

observed for root biomass, shoot length and shoot biomass (Fig. 3j–l). Taken together, our results showed that the effect of HSAL promotion on plant growth is conserved among different species.

## HSAL promotes root elongation by stimulating cell proliferation

To elucidate the underlying cellular mechanisms for the HSAL-mediated promotion of root growth, we analyzed cell numbers, cell sizes, and cell lengths in rice root tips in response to the 0.05% HSAL treatment. As shown in Supplementary Fig. 5a, the width of the rice root treated with 0.05% HSAL was increased compared to that of the seedlings grown in water without additional nutrients and HSAL treatment for 7 days. The total cell number of the root (cross-section view) at 0.05% HSAL was increased by 1.42-fold ($p = 6.8E-06$) compared to the control (Supplementary Fig. 5b). Moreover, the radial cell diameters were increased when grown in 0.05% HSAL (Supplementary Fig. 5c). However, the longitudinal cell length in the mature zone of the rice roots was not significantly changed by 0.05% HSAL treatment (Supplementary Fig. 5d). Overall, these results suggested that the increase of HSAL on the rice root length is primarily due to an effect of cell proliferation rather than increased cell length.

To test whether cell proliferation is affected by HSAL, we used the fluorescent dye 5-ethynyl-2′-deoxyuridine (EdU) that indicates the presence of DNA-synthesizing cells. Confocal microscopy of rice root meristems showed that the EdU fluorescence of the root meristem was similar between the control and 0.05% HSAL when grown in the hydroponics for 3 days. However, 8 days later, the EdU fluorescence almost vanished in the root meristems of seedlings without HSAL treatment, but was still detected at high level in the root meristems of the seedlings treated with 0.05% HSAL (Supplementary Fig. 5e–g). This suggested that 0.05% HSAL can sustain cell proliferation activity of the root meristem for a longer time, which is consistent with the difference of the root growth after 7 days of control and HSAL-treated rice plants (Fig. 3f). Together, these results support the conclusion that HSAL maintains cell proliferation and root growth in nutrient-poor conditions where otherwise cell proliferation and growth would seize.

## HSAL increases iron bioavailability to promote root growth

To pinpoint the molecular and biological processes that are targeted by HSAL and lead to enhanced root growth in nutrient poor environments, we performed a genome-wide transcript analysis of the root tip of 7-day-old seedlings treated with or without HSAL for 12 h using RNA-seq (Fig. 4a). This analysis identified 956 up-regulated genes and 1112 down-regulated genes in response to 0.05% HSAL in rice root tips (False Discovery Rate (FDR) ≤ 0.05 and Log2(Fold Change) ≥ 1). A Gene Ontology (GO) enrichment analysis identified several over-presented processes among the differentially expressed genes (DEGs) that were related to cell walls including "lignin metabolic process", "fatty acid biosynthetic process", and "carbohydrate metabolic process" (Fig. 4b, c). Interestingly, we found that several GO items related to metal nutrient transport and function such as "transmembrane transport", "oxygen transport", and "nicotianamine biosynthetic process" were also significantly enriched (Fig. 4b, c). A similar pattern of GO enrichment was found in the up-regulated genes (Supplementary Fig. 6a). In the down-regulated genes, "transmembrane transport" was the only enriched GO process (Supplementary Fig. 6c). In the Molecular Function category, "heme binding" was the most significantly enriched GO item in either all DEGs or up-regulated DEGs or down-regulated DEGs (Fig. 4b, c and Supplementary Fig. 6b, c). These results suggested that iron bioavailability including iron transport and iron related biological processes was likely modulated by HSAL treatment at the molecular level. To support this, we found that the expression of several genes involved in iron transport was significantly induced by HSAL treatment (Fig. 4d). The HSAL induced expression change of genes involved in iron transport was further confirmed by quantitative real-time polymerase chain reaction (qRT-PCR) analysis (Supplementary Fig. 6d) ($p < 0.05$). Moreover, "cation binding" was also significantly enriched in either all DEGs or up-regulated DEGs (Fig. 4c and Supplementary Fig. 6b). Collectively, the genome-wide transcriptional analysis prompted us to the hypothesize that iron bioavailability might be improved by HSAL treatment. This hypothesis was consistent with the metal chelator function demonstrated for HSAL (Fig. 2).

To test this hypothesis, we grew rice seedlings in the IRRI (International Rice Research Institute) hydroponics method[37], in iron sufficient and low iron conditions, and grew them with or without 0.05% HSAL. Treatment with 0.05% HSAL treatment increased root length by 2.84-fold ($p < 0.05$) and shoot length by 1.82-fold ($p < 0.05$) under low iron conditions (Fig. 4e–g). Notably, under low iron and HSAL treatment roots were even longer than those grown under iron sufficient conditions without HSAL (Fig. 4e, f). Consistently, iron accumulation in the root and shoot was increased 1.57-fold ($p < 0.05$) and 1.17-fold ($p < 0.05$) in 0.05% HSAL treatment under iron deficiency condition (Fig. 4h, i). These results confirmed that HSAL can improve iron bioavailability to rescue rice growth under iron deficiency conditions. Moreover, 0.05% HSAL also significantly increased the root length and shoot length of rice seedlings under iron sufficient condition (Fig. 4e–g). Iron accumulation in root and shoot was also dramatically increased 2.12-fold ($p < 0.05$) and 2.33-fold ($p < 0.05$) by 0.05% HSAL treatment under iron sufficient condition (Fig. 4h, i). These results indicate that HSAL can improve iron availability for plant growth under both iron deficiency and iron sufficient conditions. Interestingly, 0.05% HSAL also significantly increased the accumulation of calcium and copper in both the shoot and the root, while it increased the accumulation of magnesium and zinc only in the roots (Supplementary Fig. 7). These results suggest that HSAL may have an extensive capacity to improve the bioavailability of metal nutrients, especially for iron in plants.

To exclude the possibility that these effects were due to HSAL being contaminated by these nutrients, we compared the content of these nutrients in the IRRI hydroponics with and without 0.05% HSAL. Under iron sufficient condition, the concentration of iron and calcium contained in 0.05% HSAL was only 0.09% and 0.02% of the corresponding elements in IRRI medium (Supplementary Table 2). Thus, the iron and calcium contained in 0.05% HSAL were negligible for the nutrient content in the IRRI medium. Under iron deficiency conditions the pots contained 561 µg pot$^{-1}$ iron, most likely contaminated from other chemical compounds. Therefore, 1.92 µg pot$^{-1}$ additional iron from HSAL increased the total iron concentration in the growth medium by 0.05% HSAL, which does not account for the increase in total plant iron of 27.75 µg pot$^{-1}$ iron (Supplementary Table 3). Therefore, HSAL indeed increases the iron bioavailability to improve iron accumulation in rice.

## HSAL acts as a ferric iron chelator comparable to EDTA

While the monocot plant rice uses iron uptake strategy II, dicot plants use a different strategy (strategy I)[38]. We therefore tested whether HSAL could also improve iron bioavailability in the model dicot plant *Arabidopsis thaliana*. For this we used in agar medium with half Murashige and Skoog (MS) salt. Our treatment with 0.05% HSAL significantly increased chlorophyll content (1.31-fold, $p < 0.05$) and root length (3.08-fold, $p < 0.05$) of *Arabidopsis* under iron deficiency conditions (Fig. 5a–c). Additionally, we found that 0.05% HSAL also alleviated the deficiency phenotypes of other metal nutrients such as calcium and magnesium, although the recovery effect was less pronounced than that of iron deficiency (Supplementary Fig. 8). Like in rice, an increase of nutrient accumulation such as iron, calcium, magnesium, and zinc by 0.05% HSAL treatment was also found in *Arabidopsis* (Supplementary Fig. 9). Under this growth condition, the concentration of iron, calcium,

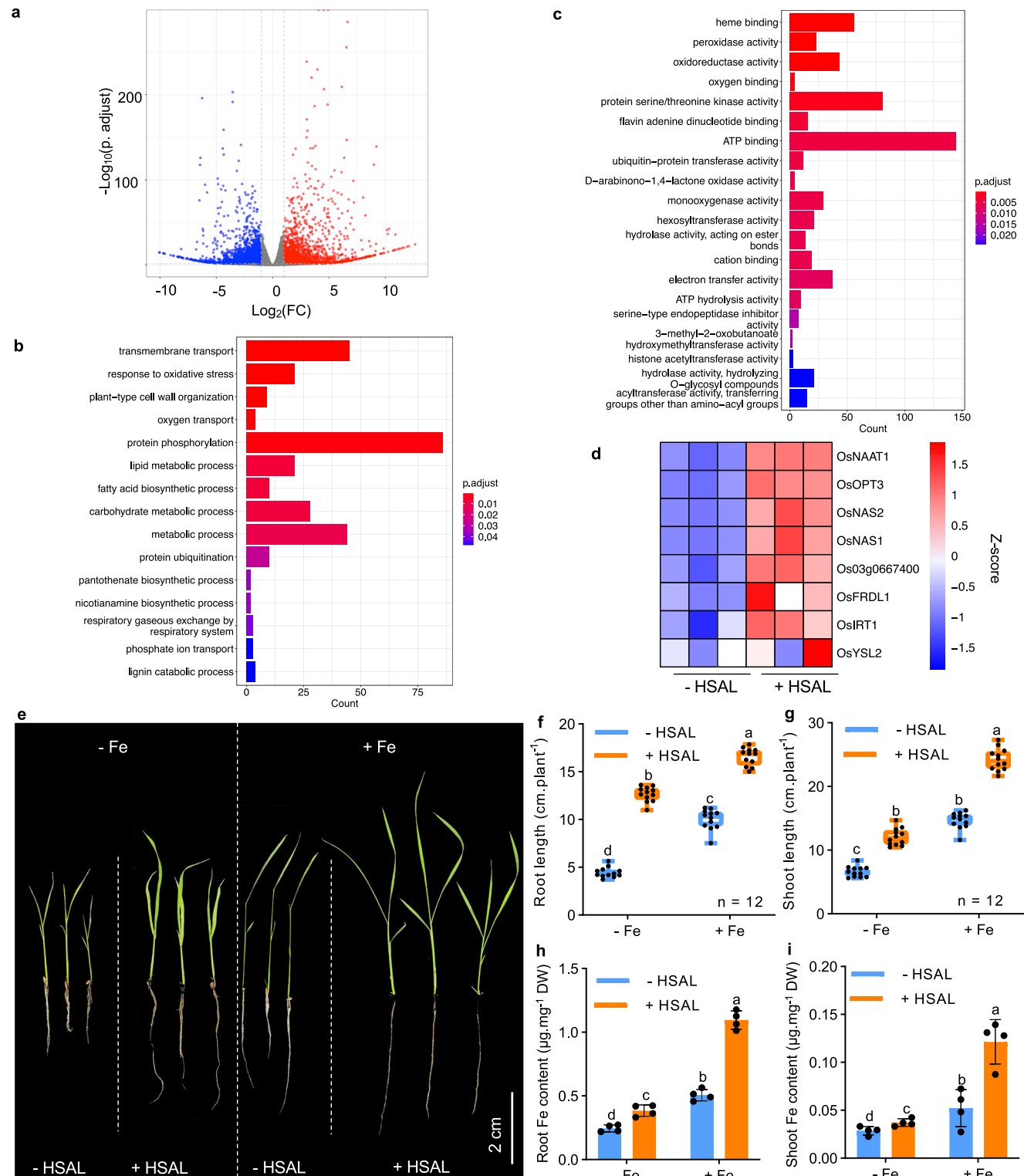

**Fig. 4 | HSAL additive significantly increases iron bioavailability for rice.**
**a** Volcano plot of 2068 significant differential expression genes (DEGs) of rice root tips treated without or with 0.05% HSAL for 12 h; significance threshold: False Discovery Rate (FDR) ≤ 0.05 and $Log_2$(Fold Change) ≥ 1. The list of DEGs was attached in Supplementary Dataset 1. **b**, **c** Significantly enriched Gene Ontology (GO) terms in Biological Process **b** and Molecular Function **c** categories of these DEGs. GO enrichment was carried out with the over representation Analysis. *p*-values were calculated by hypergeometric distribution multiple comparison followed by Benjamini-Hochberg adjustment. **d** Heat-map showing expression pattern of iron transport-related genes affected by 0.05% HSAL. **e** Representative images showing 0.05% HSAL rescues iron deficiency phenotypes in *Nipponbare*

rice. Rice seedlings were grown under iron deficiency (removed EDTA-Fe(II)) and iron sufficient (36 μM EDTA-Fe(II)), and treated with or without 0.05% HSAL for 10 days. **f**, **g** Box plots and scatter plots showing root length **f** and shoot length **g** of rice seedlings with and without 0.05% HSAL. Boxes represent the 25th–75th percentile range, white lines represent medians and whiskers show the minimum–maximum range (*n* = 12 seedlings). **h**, **i** Iron content in the roots **h** and shoots **i** of rice seedlings upon treatment with 0.05% HSAL. Graphs depict mean with standard deviation (error bars) and individual data points (*n* = 4 biological replicates). Different letters indicate statistically significant differences based on two-way ANOVA and Tukey's *HSD* test analysis (*p* < 0.05) in **f**–**i**.

magnesium, and zinc contained in 0.05% HSAL was only 0.06%, 0.01%, 0.01%, and 0.01% of the corresponding elements in half MS medium (Supplementary Table 2), so they also were negligible for the nutrient uptake by *Arabidopsis*. Consistently, the iron content of *Arabidopsis* increased upon HSAL treatment under iron deficiency condition was also not derived from HSAL-containing iron (Supplementary Table 4). Overall, these results suggested that HSAL can be an effective chelator to improve iron bioavailability and iron accumulation in *Arabidopsis thaliana*.

Different chelators can improve iron uptake from the root surface to the vascular tissue, and subsequent translocation to the shoot[39]. To examine the uptake pathways for HSAL-chelated iron in plants, we tested whether HSAL alleviated iron deficiency in *Arabidopsis* mutants in which iron uptake is impaired (Fig. 5a). These included a phloem-specific iron transporter mutant *opt3-2*[40], a plasma membrane iron transporter mutant *irt1-1*[41], and a ferric chelate reductase mutant *fro2*[42]. We found that 0.05% HSAL significantly recovered the chlorophyll content and root length of *opt3-2* to the wild-type level, while it did not recover that of *irt1-1* and *fro2* (Fig. 5b, c). These results indicate that FRO2- and IRT1-mediated iron uptake are essential for HSAL-chelated iron uptake into the plant. The plasma membrane-localized FRO2 reduces extracellular ferric chelate into ferrous iron[42], which is then transported by IRT1 via the plasma membrane into the cells[41]. Taken together, these results indicate that HSAL is a ferric iron chelator. We then compared the effect of the well-known iron chelator EDTA to the effect conferred by HSAL in all these *Arabidopsis* mutant lines (*opt3-2*, *irt1-1*, and *fro2*). The recovery pattern of EDTA on iron deficiency phenotypes in these iron uptake defective mutants was very similar to that of HSAL (Fig. 5d–g and Supplementary Fig. 10). This result further supported the model that like EDTA, HSAL can serve as a ferric iron chelator to improve iron bioavailability and rescue the plant growth under iron-deficiency conditions.

## Discussion

### The lignin-derived material HSAL serves as a sustainable iron chelator

Natural lignin is a complex phenolic organic macromolecule consisting of phenyl propane units mainly linked by carbon-oxygen-carbon bonds and carbon-carbon bonds, which require high energy to break[43]. During the process of biorefinery, lignin can be further condensed by forming new carbon-carbon bonds[44]. SAL represents an important type of industrial lignin produced in biorefineries, which is one of the most condensed lignins[22]. Using a simple hydrothermal alkaline reaction, we sheared the carbon-carbon bonds and carbon-oxygen-carbon bonds of the condensed lignin and converted SAL into the water-soluble HSAL[22]. In this study, we found a strong reduction of methoxy groups and an increment of phenolic hydroxyl groups in HSAL and characterized its metal chelating properties. Importantly, we demonstrated that HSAL acts as a sustainable nutrient chelator to promote plant metal nutrient bioavailability and growth. Although it has long been known that alkaline treatment (at different levels of alkalinity, temperatures, and pressure conditions) can dissolve lignin[44], no previous research has exploited this type of dissolved lignin from acid-alkaline conditions for plant growth promotion. HSAL can significantly promote root growth and facilitate substantially increased root and shoot biomass in both monocot and dicots plants, by improving bioavailability of iron and also other metal nutrients. HSAL improves iron bioavailability equivalent to the well-known metal chelator EDTA. Both of these chelators require FRO2 ferric reductase activity and the IRT1 transporter to facilitate the reduction of ferric iron and mediate the entry of ferrous iron into plant cells. However, due to their non-bio-degradable nature, synthetic iron chelators including EDTA-Na$_2$, and Ethydiaminedhephen Acetic-Na (EDDHA-Na) will accumulate in the soil and are associated with a series of secondary risks such as heavy metal overload in water and crops[45–47]. In contrast, HSAL will likely be degraded similarly to the structurally related humic substances in soils[48]. Numerous studies have also shown plant growth promotion of other industrial lignin-derived materials such as lignosulfonates that is produced as a byproduct of the manufacturing of paper pulp through the sulfite process and lignin-based nanoparticles, although the mechanisms underlying this are not well studied[49,50]. Therefore, HSAL represents a novel metal chelator that can be produced at a lower cost and in a sustainable manner as it prevents greenhouse gas emissions from burning waste lignin. It is therefore a promising compound for replacing synthetic chelating agents in future.

### HSAL enhances plant nutrient bioavailability through its metal chelating capacity

As shown in the model of HSAL function for improving plant growth in Fig. 6, we propose two major steps for HSAL to increase iron bioavailability through metal chelating capacity. Firstly, HSAL could chelate ferric iron in the medium to improve iron mobility, which increases iron diffusion to the root apoplastic space and thus improves iron bioavailability. Secondly, HSAL could chelate apoplastic ferric iron to avoid iron precipitating in cell walls and thus improve iron bioavailability. The latter mechanism is supported by the result that HSAL-dependent iron entry into plant cells requires FRO2 ferric reductase activity and the ferrous IRT1 transporter in Strategy I nongraminaceous plant *Arabidopsis*. In Strategy II plants like rice, roots take up iron in the form of phytosiderophore chelated ferric iron complexes. However, we do not think HSAL chelated ferric iron directly enters into plant cells, because the molecular weight of HSAL is higher than 3500 (we had collected HSAL using a cellulose tube with a 3500 molecular weight cutoff), and natural lignin-derived substances with a high molecular size fraction (Mw > 3500) has been suggested to be not directly absorbed by roots[51]. Alternatively, ferric iron might be released from HSAL chelated complexes, and transported into root cells in the form of phytosiderophore chelated complexes.

Application of HSAL also rescued the growth defect of *Arabidopsis thaliana* caused by the deficiency of different nutrients, including Ca, Mg, Zn, and Cu (Supplementary Fig. 8), and also increased the total amounts of these metal nutrients especially Ca in plant tissues (Supplementary Fig. 9). These results therefore indicate that the function of HSAL is not specific to iron, which is consistent with the metal chelating capacity of HSAL. We think HSAL most likely improves the mobility of these ions in the growth medium to increase metal nutrient diffusion to the root surface and/or to avoid iron precipitating in cell walls through its chelating capacity, which leads to increased bioavailability of these metal nutrients for plants.

### Economic assessment of HSAL

Firstly, the conditions and cost for HSAL production are relatively simple and low. HSAL is a medium sized lignin material (around 7.5KD) that is converted from the SAL prepared in the laboratory with an alkaline (NaOH solution) hydrothermal reaction at 280 °C for 2 h. If the waste lignin SAL produced in the biomass refining industry is used for HSAL production, it may increase the cost as additional procedures such as purification and isolation of the original industrial wastes are required, but it might reduce the amount of subsequently NaOH solution due to additional acid-base neutralization consumption. The cost of HSAL production was calculated at around USD 329–430 per ton based on the laboratory conversion, and can be further reduced to USD 139–213 per ton by the large-scale industrial production (see methods section for details). This price is far lower than USD 3500–350,000 per ton for synthetic metal chelating agents including EDTA, EDDHA, and DTPA[52]. The cost of HSAL production is also much less than USD 495–850 per ton for lignosulfonate[53]. The production process and cost for lignin-based nanoparticles are more complicated

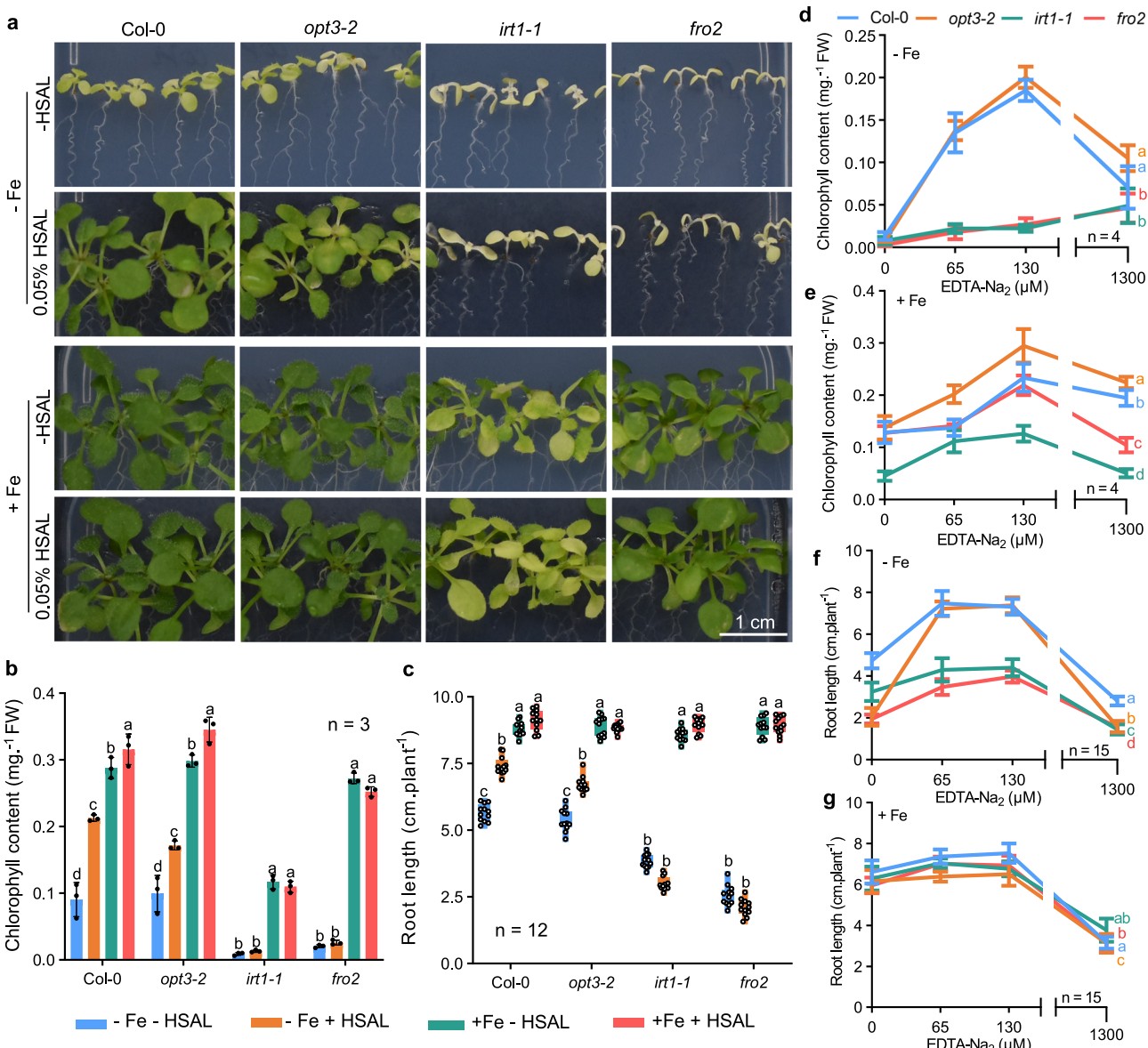

**Fig. 5 | HSAL mediated increase of iron bioavailability is comparable to that mediated by EDTA. a** Representative images showing the effect of 0.05% HSAL on the recovery of iron deficiency phenotype in different *Arabidopsis* iron uptake defective mutants. Seedlings of wild-type Col-0 and mutants *opt3-2*, *irt1–1* and *fro2* were grown under ½ MS medium (+Fe) and iron-deficient medium (-Fe) for 10 days. **b** Total chlorophyll content. Graphs depict mean with standard deviation (error bars) and individual data points (*n* = 3 biologically independent replicates). **c** Box plots and scatter plots showing root length. Boxes represent the 25th–75th percentile range, white lines represent medians and whiskers show the minimum–maximum range. Different letters indicate statistically significant differences based on one-way ANOVA and Tukey's *HSD* test analysis (*p* < 0.05) in **b–e** Effect of different concentrations of EDTA-Na₂ (0–1300 μM) on the recovery of iron deficiency phenotypes in wild-type *Arabidopsis* and different iron uptake and transport defective mutants (*opt3-2*, *irt1–1*and *fro2*). The seedlings were grown on iron-sufficient medium (½ MS with 50 μM EDTA-Fe (III), pH = 5.7) or iron-deficient medium (½ MS without EDTA-Fe (III), pH = 5.7) for 9 days. Total chlorophyll content under iron deficiency **d** and iron sufficient conditions **e**. *N* = 4 biologically independent replicates. **f**, **g** Root length under iron deficiency **f** and iron sufficient conditions **g**. The lines indicate mean and standard deviation (error bars). *N* = 15 seedlings. Different letters indicate statistically significant differences among genotypes **d–g** based on one-way ANOVA and Tukey's *HSD* test analysis (*p* < 0.05).

and higher[54]. Secondly, there is a potentially very high demand of HSAL production to increase metal nutrients and growth for major staple crops. For example, one-third (around 0.52 billion hectares) of global arable land is iron-deficient soil[55], which requires a large amount of iron chelator fertilizer additives to improve the yield and quality of crops[46]. Potentially, up to 17.9 million tons of HSAL could be applied per year to this iron-deficient arable land at a concentration of 0.05%[55]. Based on the HSAL conversion rate of 46.7% (Fig. 1), it would require 38.2 million tons of lignin, which would consume 76.6% of the total lignin (50 million tons) of the industrial waste discharged by lignocellulosic

biorefinery industry in 2020[1]. Thirdly, application of HSAL not only reduces carbon emissions from biorefinery factories but also promises to increase soil carbon storage. If most industrial lignin would be converted into HSAL and applied to iron-deficient soils, it would reduce 116.8 million tons of $CO_2$ emissions by avoiding burning of the source material[56]. Moreover, the increase of root growth of plants could enhance carbon sequestration in soils[57–59]. However, proper approaches to improve depth and biomass of the crop root systems have not been developed yet. The beneficial effects of HSAL might provide a promising way to contribute to achieving this goal.

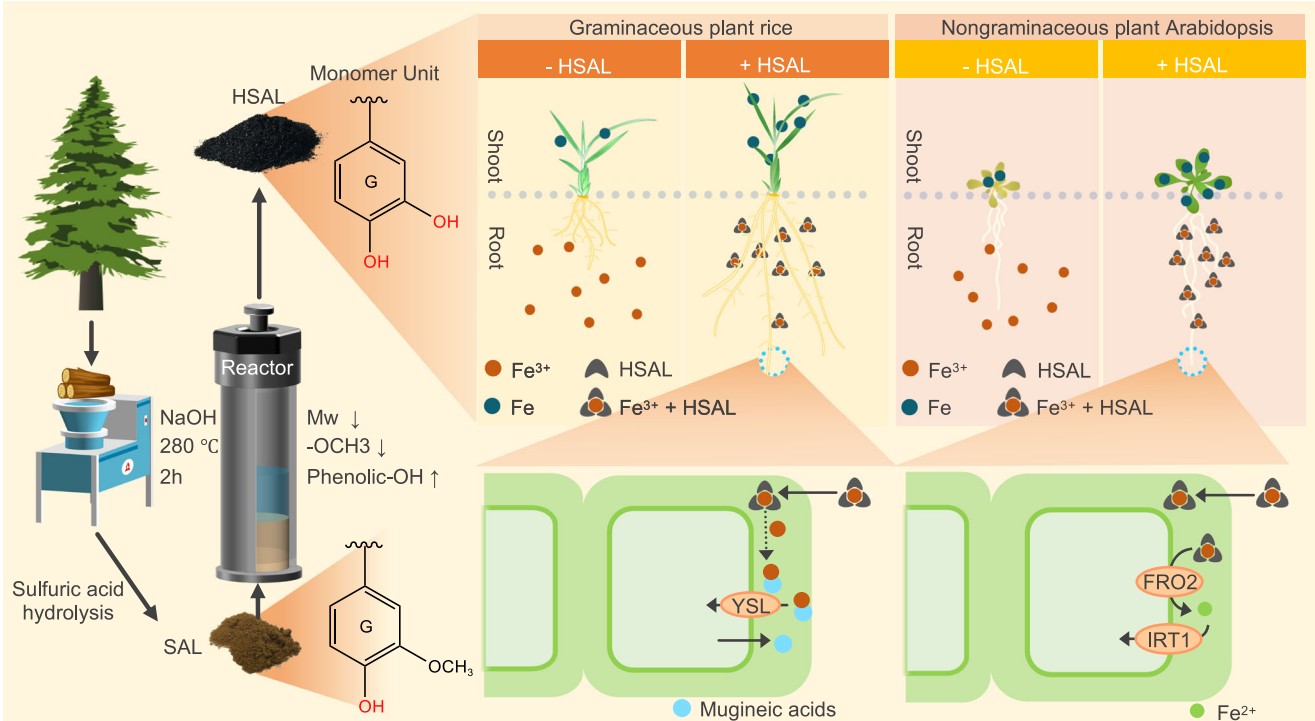

**Fig. 6 | Schematic diagram for HSAL synthesis, structure and its capacity for enhancing iron bioavailability to increase root length and plant biomass.** Sulfuric acid lignin (SAL)is converted into a water-soluble lignin-based material (HSAL) through a hydrothermal reaction. This reaction not only shears the bonds of the monomers to depolymerize the lignin, but also causes the decrease of methoxy groups and increase of phenolic hydroxyl groups. This changed structure makes HSAL to act as a sustainable ferric iron chelator promoting plant growth and root length by chelating ferric iron to enhance iron bioavailability in the growth medium and apoplastic space in both rice and *Arabidopsis*. In the Strategy I plant *Arabidopsis*, HSAL chelated ferric iron is reduced by the FRO2 ferric reductase into ferrous iron and is then transported into the plant cells through the IRT1-transporter. In the Strategy II plant rice, ferric iron might be released from HSAL chelated complexes, and transport into root cells through YSL transporters in the form of phytosiderophore chelated complexes.

## Methods

### HSAL synthesis

To obtain high-purity sulfuric acid lignin (SAL) that simulates the lignin byproduct from lignocellulosic biorefinery by using concentrated sulfuric acid, we used the Klason method to fully remove cellulose and other carbohydrates in wood powder by using concentrated sulfuric acid hydrolysis. A total of 12.5 g wood powder (60–100 mesh) of Japanese cedar (*Cryptomeria japonica*) was extracted with the Soxhlet solution (Benzene/ethanol = 2:1) and followed application of a Klason method (TAPPI Standard T-222, 2006)[22]. Briefly, 72% concentrated sulfuric acid was added to the solution and incubated at room temperature for 30 min; then the sulfuric acid was diluted to 3% with water and the reaction was performed in a 90 °C-oil bath for 90 mins, the resulting insoluble substrate was SAL. After that, a mixture containing 500 mg of SAL, 500 mg of NaOH, and 15 mL of water was sealed in a stainless-steel tube with an inner volume of 50 cm³ and heated at 280 °C for 2 h. After the reaction, the stainless-steel tube was cooled immediately to room temperature by immersion in a water bath. The reaction mixture was filtered using a 1G3 filter (Sekiya, Tokyo, Japan) to obtain the liquid form of the hydrothermally treated SAL (called HSAL). The HSAL solution was collected carefully and dialyzed using a cellulose tube with a 3500 molecular weight cutoff (Tubing TM 3500; Bio-Design, Carmel, NY, USA) to remove inorganic reagents. HSAL powder was obtained using a lyophilization process. The solubility of HSAL was determined by the amount dissolved in 100 g of ultrapure water when it reached saturation at room temperature (25 °C)[60].

### Characterization of HSAL chemical structures

The overall differences in functional groups of SAL and HSAL were determined by Fourier transform infrared spectroscopy (FT-IR)

spectroscopic analysis using a SHIMADZU8400s (Kyoto, Japan) over the region between 400 and 4000 cm$^{-1}$ with a resolution of 4 cm$^{-1}$ and 32 scans. Each sample was prepared using the potassium bromide method[61]. The methoxy content of SAL or HSAL was measured using a modified TAPPI method (TAPPI Standard T209 su-72, 2016). Briefly, 3 mg SAL or HSAL reacted with hydroiodic acid (57%, w/w) in a sealed glass vial at 130 °C for 20 min to generate methyl iodide. The methyl iodide content was determined by a Gas Chromatography-Flame Ionization Detector (GC-FID, GL Sciences, Tokyo, Japan) with a TC-1 capillary column (60 m × 0.25 mm i.d.; GL Sciences) with an N$_2$ carrier flow. Relative quantitative comparison of methoxy content was based on methyl iodide determination of SAL or HSAL (Ethyl iodide dissolved in tetrachloromethane solution was used as a standard solution).

The total amount of free hydroxyl groups in SAL or HSAL was determined using the acetylation method[62]. 0.2 g of SAL or HSAL reacted with the acetylation reagent (containing 1 ml acetic anhydride and 1 ml pyridine) at 50 °C for 48 h, then 2 ml acetone and 2 ml water were added, and then 0.01 M NaOH solution was used to determine the acetic acid content by potentiometric titration. The total free hydroxyl group was calculated based on the consumption of 0.01 M NaOH solution. The distribution of phenolic hydroxyl groups and alcoholic hydroxyl groups of HSAL was characterized based on the amount of aliphatic and phenolic acetate protons recorded using ¹H NMR spectra (Bruker AVANCE 400 MHz, Germany) of acetylated HSAL. Quantification of the aliphatic acetates (1.6–2.1 ppm) and phenolic acetates (2.1 ppm to 2.6 ppm) was achieved by peak integration in comparison with dibromomethane internal standard peaks (4.9 ppm). As SAL appears to be almost insoluble in NMR organic reagents and difficult to be detected by NMR methods, the phenolic hydroxyl content of SAL was determined by using the

selective ammonolysis method[62]. Briefly, the hydroxyl group of SAL was first acetylated by an acetylation reagent (containing 1 ml acetic anhydride and 1 ml pyridine) and stirred at room temperature under nitrogen protection for 48 h, then a known amount of internal standard N-propionyl pyrrolidine (0.1 M) was added to a pyrrolidine solution for acetyl group ammonolysis. The amount of N-acetylpyrrolidine generated at the beginning of the reaction was determined by a gas chromatography analysis (Shimadzu GCMS-QP 2010) and used to obtain the phenolic hydroxyl content of SAL. The amount of methoxy groups, phenolic hydroxyl groups, and alcoholic hydroxyl groups per 100 C9-units was calculated based on the quantitative analysis of those functional groups and the degree of polymerization in lignin that was converted from the average molecular weight of SAL or HSAL.

## Analysis of HSAL chelating capacity

To obtain HSAL-metal chelating complexes, 0.2 g HSAL was dissolved in 20 ml of ultrapure water, and 0.1 M of $FeSO_4$ (15.19 g), $FeCl_3$ (16.22 g), or $CaCl_2$ (11.09 g) respectively were added. This was then stirred at room temperature for 6 h. The liquid was introduced into the cellulose membrane, rinsed in water for 96 h, and freeze-dried (FM25EL, SP Industries, USA) to collect the different HSAL chelating complexes. The element composition and element valence state of HSAL and its chelating complexes were analyzed using an X-ray photoelectron spectrometer (XPS, ESCALAB MARK II, VG Instruments, UK) with monochromatized Al Kα (hυ = 1253.6 eV). The spectrometer was calibrated by the binding energy (50.0 eV) of Au 4f7/2 with reference to the Fermi level. The samples were mounted on a conductive carbon tape held in a copper sample holder and transferred directly into the analysis chamber. The XPS lines were calibrated using the C 1 s peak at 284.8 eV that was present at the surface of all samples. The model peak to describe XPS core-level lines for curve fitting was a product of Gaussian and Lorentzian functions. The functional group analysis of different HSAL chelating complexes was performed using FT-IR spectroscopy (Vertex 70, Bruker, Germany). The region between 4000 $cm^{-1}$ and 800 $cm^{-1}$ was recorded with a resolution of 4 $cm^{-1}$ and 32 scans using KBr pellets[61].

To test whether crystalline hybrid materials formed in HSAL and HSAL chelated with $FeCl_3$, X-ray powder diffraction analysis (XRPD) was performed using a Escalab 250Xi diffractometer (Thermo Fisher, UK) with Cu Kα radiation equipped with a Si(Li) solid-state detector. The recorded angular range was from 5° to 70° (2θ) with a scanning width of 0.05° in 2 s per step[34]. Nicolet 5PCFT-IR spectrophotometer equipped with ATR accessory (diamond) was conducted for the infrared spectra at range of 400–4000 $cm^{-1}$. To further observe the micromorphological differences and element mapping of HSAL and HSAL chelated with $FeCl_3$, a Tecnai G2 F20 S-TWIN (FEI, USA) transmission electron microscope (TEM) operating at 200 kV with an Oxford X-Max$^N$ 80 T energy dispersive X-ray spectrometer (EDS) detector was used to characterized atomic composition at specific positions[35]. The specimens were prepared by grinding the powders and drop casting on holey carbon film.

To quantify the iron chelation value of HSAL and EDTA, 0.1 g of HSAL or EDTA was dissolved in 10 ml of distilled water with an indicator (5% sulfosalicylic acid), and titrated with 0.01 M $NH_4Fe(SO_4)_2$ standard solution. The titration endpoint was indicated by the color of the solution changing from yellow to light red. To determine the calcium chelation value of HSAL and EDTA, 0.1 g of HSAL or EDTA was dissolved in 10 ml of distilled water, supplemented with 1 ml of ammonia-ammonium chloride buffer solution composed of 1 M ammonium chloride and 9.8% ammonia at pH 10, and an indicator (acid germanium blue K: naphthol green B: potassium chloride = 1:2:40), and then titrated with 0.2 M calcium acetate solution. The titration endpoint was indicated by the color of the solution changing

from bright blue to purple red. The $Ca^{2+}$ and $Fe^{3+}$ chelation values of HSAL and EDTA were calculated based on the titration consumption.

## Plant materials

Seedlings of three different varieties of *Oryza sativa L. japonica* rice (*Nipponbare*, *Taichung 65*, and *Kasalath*); Corn (*Zea mays L.*), *Arabidopsis thaliana* accession Columbia-0 (Col-0) and mutants *irt1-1*[41], *frd1-1* (*fro2*)[42] and *opt3-2*[40] were used in this study.

## Growth conditions and phenotypic analysis

Rice seeds were sterilized by soaking in a Benlate solution (2.5 g $L^{-1}$) and vernalization at 28 °C for 3 days. These rice seedlings were transferred into a hydroponic growth system with different concentrations of HSAL (0%, 0.01%, 0.05%, and 0.1%) dissolved in filtered water without additional nutrients in an opaque plastic box, and grown in a growth chamber (MLR351, Sanyo, Japan) at 28 °C long-day condition (16 h light 16000 Lux 28 °C/8 h dark 25 °C) for 14 days. Images of rice seedlings (*n* = 12) were taken with a digital camera (D90, Nikon, Japan). The root length and shoot length of the rice seedlings were measured with a ruler. Dry weight of plant tissue was used to measure biomass.

For rice iron deficiency experiments, rice seeds were immersed twice for 10 min in a 10% hydrogen peroxide (7722-84-1, SCR, China) solution and sterilized with 10% sodium hypochlorite (7681-52-9, SCR, China) for 15 min with agitation. These rice seedlings were vernalized at 28 °C for 3 days before being transferred into the iron deficiency (without adding EDTA-Fe(II)) or iron sufficient (36 μM EDTA-Fe(II)) IRRI hydroponic solution (hydroponic nutrient solution was based on the recipe of International Rice Research Institute) with or without 0.05% HSAL in an opaque plastic box, and grown in a growth chamber (GXZ-500C-3, Jiangnan, China) with a long-day condition (16 h light 16000 Lux 28 °C/ 8 h dark 25 °C) for 10 days. Root length and shoot length of the rice seedlings (*n* = 12) were measured with a ruler. Roots and stems were sampled separately and dried for subsequent elemental analysis.

Corn (*Zea mays L.*) seeds were sterilized by treatment with a diluted $H_2O_2$ solution. Sterilized seeds were germinated on a moist filter paper in the dark at 25 °C for 4 days. Corn seedlings were then transferred to grow in soil collected from Higashiyama Campus of Nagoya University (GPS coordinate: 35°10'00.00"N, 136°55'00.00"E), placed in a growth chamber supplying 8 h light 16000 Lux 19 °C/16 h dark 17 °C for 2 weeks. Corn seedlings (*n* = 11) were watered with filtered water (control), 0.4% fertilizer (HYPONex, diluted 250 times) (fertilizer group), 0.4% of HSAL (HSAL group), or a combination of 0.4% fertilizer, and 0.4% of HSAL (fertilizer + HSAL group), respectively, once (200 ml) every 2 days. After 14 days of planting, the roots were dug out and rinsed with water. Root length was measured with a soft ruler, and dry weight of plant tissue was measured to obtain the biomass.

*Arabidopsis* seeds were surface sterilized using 1% sodium hypochlorite disinfectant and vernalized at 4 °C for 2 days. These seeds were then sown on the plates containing ½ Murashige and Skoog (MS) salt with 1% agar. The plates were vertically placed in a growth chamber (GXZ-380C-4, Jiangnan, China) at 22 °C with 16 h light /8 h dark cycles. For HSAL treatment, a concentration of 0.05%(m/m) HSAL was added to ½ MS medium. For nutrient deficiency experiments, the corresponding nutrient as indicated in the figures was removed from the ½ MS salt. For EDTA treatment, different concentrations of EDTA-$Na_2$ as indicated in the figure were added into the ½ MS medium with or without iron deficiency. The images of the seedlings (*n* = 12) were taken with a camera (Nikon, D5600, Japan) when grown for 10 days. The root length was analyzed with Image J (https://imagej.net/imagej-wiki-static/Fiji).

## Cell morphology analysis

Rice seedlings were grown in water with or without 0.05% HSAL for 7 days. The main root segments (1.5 mm to 8.5 mm from root tip) of these seedlings were collected and dehydrated using 30%, 50%, 70%, 95%, and 100% ethanol series for 30 min each step. 100% ethanol was changed twice and the tissue was stored overnight at 4 °C and embedded in epoxy resin (EPON812; TAAB, Berkshire, UK). Transverse sections of approximately 1 μm thickness were cut with a glass knife on an ultramicrotome (Ultracut N; Reichert, Vienna, Austria) for the Cryo-SEM observation. The frozen sample was fixed in a sample holder, cut in the glove box (below −30 °C) under a dry $N_2$ atmosphere, and transferred to the cryo-scanning electron microscope (Cryo-SEM, S-3400N-T3, Hitachi, Tokyo, Japan) through cryo-vacuum transfer shuttles (TRIFT III, ULVAC-PHI Inc., Kanagawa, Japan). The cell morphology was observed with Cryo-SEM[63].

## Cell proliferation analysis

Rice seedlings were cultivated in water for 5 days, then transferred into 0.05% HSAL solution for 1.5 h, 3 h, 6 h, 12 h, 24 h, 72 h, and 192 h in the growth chamber. EdU fluorescence staining was used to indicate the cell proliferation in the root tip[64]. EdU staining (ThermoFisher, A10044) was conducted following the manufacturer's protocols. In brief, rice seedlings were immersed with 5 μM EdU for 2 h. The root tip up to 5 mm of these EdU-stained roots were cut and fixed with 4% FAA and vacuum infiltrated for 10 min. After membrane saturation and reacting with reaction cocktail, samples were stained with DAPI at room temperature for 20 min. EdU fluorescence for each root sample was observed using a confocal laser scanning microscope (FV1200, Olympus, Japan) at 454 nm after dehydration through stepped concentrations of ethanol and MES (Methyl salicylate).

## Genome-wide transcriptomic analysis

Rice seedlings (Oryza sativa L. Nipponbare) were cultivated in water for 7 days. Root tips (around 5 mm) of rice seedlings (28 root tips per group) with or without HSAL treatment for 12 h. We extracted the total RNA of the root tips with the TRIzol reagent (Invitrogen) by following the manufacturer's instructions. The total RNA was subject to library preparation with Truseq stranded mRNA sequencing kit (Illumina Inc.) and then subjected to single-end sequencing with the Illumina Nova-seq 6000 platform. The raw reads were filtered to obtain high-quality reads without adaptor sequences and low-quality reads (containing over 10% of N or trim the ends of reads with low sequencing quality (<Q20)). After quality control, only the clean reads were used for downstream analysis. The reference genome and gene model annotation files were downloaded from the genome website (https://plants.ensembl.org/). The clean reads were aligned to the Oryza sativa L. Nipponbare genome assembly (version 4) using Tophat (v2.0.9)[65]. Gene expression levels were calculated using the FPKM (fragments per kilobase of transcript per million mapped reads) metric using the Python package HTSeq with default settings. The DESeq2 R (1.10.1) package[66] was used to evaluate differential gene expression between two groups. The resulting $P$ values were adjusted using Benjamini and Hochberg's approach for controlling false discovery rate (FDR). Differential expression genes were determined using the criteria of FDR ≤ 0.05 and fold change ($log_2FC$) ≥ 1 and provided in Supplementary Data 1. To conduct Gene ontology (GO) enrichment analysis, the universal GO term annotations were retrieved from the R package GO.db (v 3.15.0). The Rice GO dataset was downloaded from AgriGO v2 (http://systemsbiology.cpolar.cn/agriGOv2/). Then a domestic GO ID/Gene ID as well as a GO ID/GO annotation file was prepared. GO enrichment analysis was carried out using the enricher () function from the R package clusterProfiler (v 4.4.4)[67] with Benjamini and Hochberg's approach for $P$ value adjustment. GO terms with adjusted $P$ values lower than 0.05 were plotted.

## Quantitative real-time polymerase chain reaction (qRT-PCR)

Rice seedlings (Oryza sativa L. Nipponbare) were cultivated in water for 7 days. The root tips (around 5 mm) of rice seedlings (12 root tips as a group) with or without HSAL treatment for 12 h were collected for RNA isolation. Six biological replicates were analyzed for each treatment. Total mRNA of the root tips was extracted using FastPure Plant Total RNA Isolation Kit (code no.RC401-01, Vazyme, China). A total of 150 ng RNA was used to synthesize the first-strand cDNA by HiScript II Q Select RT SuperMix for qRT-PCR (+gDNA wiper) (code no. R223-01, Vazyme, China). Pro Universal SYBR qPCR Master Mix (code no. Q712-02, Vazyme, China) was used to prepare the reaction mixture. PCR was performed on a StepOnePlus Real-Time PCR System (code no. 4376600, Thermo Fisher Scientific, Carlsbad, CA, United States). The primers for qRT-PCR were listed in Supplementary data 2 (Sangon Biotech, China). The specificity of primers was confirmed by melting curve analysis. eLF-4α was used as an internal reference.

## Chlorophyll content analysis

Approximately 20 mg fresh leaves of 10-day-old Arabidopsis seedlings were frozen with liquid nitrogen and ground up (Scientz-48 Touch Display High Throughput Tissuelyser, Ningbo, China) with steel beads in 2 ml tubes, then 1.8 ml of extraction buffer was added (ethanol: acetone: water = 4.5:4.5:1; v/v) and shaken (Orbital shaker TS-3D, Kylin-Bell, China) in the dark at room temperature for 15 mins. 1 ml supernatant was used to measure chlorophyll content after centrifuging the samples at 6000 rpm for 30 s. The absorbance of chlorophyll was measured at wavelengths 645 nm and 663 nm using an ultraviolet spectrophotometer (The SpectraMax® i3x Multi-Mode Microplate Reader, Molecular Devices, LLC.). Total chlorophyll content was calculated with the formula: $(20.29*A645 + 8.05*A663) *v/w*1000$[68].

## Elemental analysis

HSAL powder (20 mg) was digested with 4 mL $HNO_3$ and 2 mL $H_2O_2$ using a microwave digestion system (MARS6, CEM Microwave Technology Ltd., USA) until the liquid was clear and transparent. This was diluted with ultrapure water to a final volume of 10 mL. The contents of K, Ca, Mg, Fe, Cu, and Zn in the digestion liquid were detected using inductively coupled plasma-mass spectrometry (ICP-MS Nexion300xx, Perkin Elmer, Inc., USA). The 10-day-old seedlings of Arabidopsis thaliana and rice (Oryza sativa L. japonica. Nipponbare) were washed using the cleaning solution (10 mM EDTA-Na₂) for 30 min and subsequently rinsed with ultrapure water, then divided into roots and shoots. The samples were dried at 70 °C for 48 h and weighed. The dried samples were incubated with 2 mL of concentrated $HNO_3$ under high-temperature (max to 220 °C for Arabidopsis thaliana and 280 °C for rice) until the liquid was clear and transparent. The samples were then diluted with ultrapure water to a final volume of 10 mL. The contents of Ca, Mg, Fe, Cu, and Zn in the digestion liquid were detected using a Microwave Plasma-Atomic Emission Spectrometer (4210-MP-AES; Agilent Technologies, USA).

## Cost assessment of HSAL synthesis

The cost calculation of HSAL conversion was based on the current experimental conditions: 3% SAL (water base) treated with the same mass of NaOH at 280 °C for 2 h, and the yield of SAL converted to HSAL by hydrothermal reaction is 46.7%. Accordingly, the daily production cost[69] of one ton of HSAL was determined by considering the energy consumption of the corresponding SAL-NaOH mixed liquid (specific heat capacity of 3% NaOH (c): 0.95 kcal. $(kg °C)^{-1}$) hydrothermal treatment (converted into standard coal $(2.93 × 10^4 kJ kg^{-1})$ equivalent, USD 107–122 per ton[70]) and the material consumption caused by NaOH addition (recycling, calculated at 10% loss each time, USD 300–600 per ton (Dadao Chemicals Co., Ltd.)). In the scenario of large-scale

industrial production, the additive concentration of SAL would be increased to 10% (specific heat capacity of 10% NaOH (c): 0.89 kcal. (kg °C)$^{-1}$) to further improve production efficiency and cut costs. Under the premise that the treatment conditions of the hydrothermal reaction are consistent, the daily production cost of one ton of HSAL was calculated.

$$
\text{Daily production cost of 1t HSAL} = \left\{ \begin{array}{l} [(c(SAL-NaOH)*m(SAL-NaOH)*(T2-T1)/Q(\text{Std. Coal})]*P(\text{Std. coal}) \\ +10\%*m(SAL)*P(NaOH) \end{array} \right\} / \text{Yeild of HSAL}
$$

c(SAL-NaOH): Specific heat capacity of the corresponding SAL-NaOH mixed liquid;

m(SAL-NaOH): Total mass of the corresponding SAL-NaOH mixed liquid;

T2: Target temperature of hydrothermal reaction (°C);

T1: Start temperature of hydrothermal reaction (°C);

Q (Std.Coal): Calorific value of standard coal ($2.93 \times 10^4$ kJ kg$^{-1}$);

P (Std.Coal): Average market price of standard coal;

m (SAL): Mass of the invested SAL (assumed as 1 Ton here);

P (NaOH): Average market price of NaOH;

Yield of HSAL: Yield of HSAL produced by hydrothermal conversion (Calculated at a laboratory yield of 46.7%).

### Quantification and statistical analysis

All values were presented as means ± standard deviation (SD) and box plots. The experiments were performed at least for three biological replicates (the number of samples for each value was indicated in figures or figure legends) and repeated twice. For box plots, boxes represent the 25th–75th percentile range, white lines represent medians and whiskers show the minimum–maximum range. Statistical analyses between groups were analyzed with Student's $t$-test ($p < 0.05$) using the GraphPad Prism 9 (GraphPad Software) for Windows. Significant differences for multiple comparisons for single point experiment were determined by one-way or two-way ANOVA with Tukey's *HSD* test as indicated in figure legends. All statistical tests used were two-sided in this study.

### Reporting summary

Further information on research design is available in the Nature Portfolio Reporting Summary linked to this article.

## Data availability

The data that support this study are available within the paper and Supplementary Information files. The data underlying Figs.1b-f, 2b-f, 3b-g, 3i-l, 4b-d, 4f-i, 5b-g and Supplementary Figs. 2a, 3b-g, 4b-g, 4b-d, 5b-d, 5f-g, 6a-d, 7a-h, 9a-h, as well as Supplementary Table 1–4 generated in this study are provided in the Source Data file. The raw RNA-seq datasets of *Oryza sativa* used in this study are available in the GEO database under accession code GSE224234. Source data are provided with this paper.

## Code availability

All codes used in this study are openly available at Zenodo (https://doi.org/10.5281/zenodo.7912947)[71].

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

## Acknowledgements

We thank Xuhai Zhu (Dalian Institute of Chemical Physics, Chinese Academy of Sciences), Xiao Jiang (North Carolina State University) and other members from Li lab and Matsushita Lab for valuable discussions and comments; Guoqing Zhu (Center of Electron Microscopy, Zhejiang University) for assistance with TEM analysis; Lijuan Mao (Analysis Center of Agrobiology and Environmental Sciences, Faculty of Agriculture, Life and Environment Sciences, Zhejiang University) for FT-IR analysis; Jianli Shen (Experimental Center of Materials Science and Engineering, Zhejiang University) for XPRD analysis; Hanjie Zhang (Institute of Condensed Matter Physics of Zhejiang University) for XPS analysis. This work is supported by the National Natural Science Foundation of China (Nos. 32172656 and U21A20234, to B.L.), the Natural Science Foundation of Zhejiang Province (No. LZ22C020002, to B.L.), the Fundamental Research Funds for the Central Universities (226-2022-00084, to B.L.), the Japan Society for the Promotion of Science KAKENHI (grant no. 18H03959, to Y.M.) and by Emachu-foundation (to Y.M.).

## Author contributions

Q.L., Y.I., Y.M. and B.L. conceived and designed the projects. Q.L., Y.M., T.K. and D.A. performed the experiments, Q.L., T.K., Y.I., D.A., Z.F., Y.X., Y.M. and B.L. analyzed the data. Y.I., K.F., X.L., Y.M. and B.L. contributed materials and tools. Q.L., W.S., W.B., Y.M. and B.L. wrote the paper with contributions of all authors.

## Competing interests

The authors declare no competing interests.
