## [Peer Review File · Nature Communications]

REVIEWER COMMENTS

Reviewer #1 (Remarks to the Author):

In this manuscript, Liu et al. nicely demonstrated that the water soluble hydrothermal sulfuric acid lignin (HSAL) can serve as an eco-friendly and economical nutrient chelator to improve plant nutrient bioavailability and enhance plant growth. This manuscript covers several aspects involved in material science, molecular science and plant nutrition, from the synthesis and characterization of HSAL to physiological roles of HSAL in promoting plant growth and the underlying molecular mechanism. I am not familiar with the part of the synthesis and characterization of HSAL but really enjoy in reading how they demonstrated HSAL functioning as an iron nutrient chelator to promote plant growth. By adding 0.05% HSAL, the authors nicely showed that HSAL did promote growth of several plant species. Taking the advantage of genome-wide transcriptomic analysis they found that HSAL mainly acts as a ferric iron chelator to increase iron bioavailability and thus promote plant growth under both iron limited or sufficient conditions. This conclusion was elegantly confirmed by using several Arabidopsis mutants, including the iron uptake transporter mutant *irt1*, the ferric chelate reductase mutant *fro2* and the phloem-specific iron transporter mutant *opt3*. Overall, this manuscript provided a sustainable route to utilize industrial lignin by conversion to HSAL which can potentially replace synthetic chelating agents and serve as a promising eco-friendly metal chelator to promote the profitability and sustainability of biorefineries and reduce the immission of CO₂ emissions. The experiments in the manuscript were well designed and the conclusion was convincing. I don't have any critical comments but a few suggestions as below.

1. Supplementation of 0.05% HSAL promotes the growth of both graminaceous plants (rice and maize) and nongraminaceous plant (*Arabidopsis thaliana*). These two type plants use different Fe acquisition strategies with Strategy I in nongraminaceous plants and Strategy II in graminaceous plants. It is reasonable that HSAL addition could promote the growth of *Arabidopsis*, the Strategy I plant, as HSAL-chelated Fe(III) can be reduced to Fe(II) and uptake by IRT1. This is also evidenced by using the *Arabidopsis irt1* and *fro3* mutants. For the Strategy II plants like rice, plants take up iron in the form of phytosiderophore chelated Fe(III), for example, deoxymugineic acid chelated Fe(III) for rice. So is it possible Strategy II plants could direct uptake the HSAL chelated Fe(III)? I would suggest to discuss this point in the Discussion.
2. In Fig 6, rice, a Strategy II plant, was used on the top right of the schematic diagram but the Strategy I iron acquisition pathway was showed in the bottom right. This inconsistency must be corrected.
3. Numerous differentially expression genes (DEGs) were identified in the root tip of 7-day-old seedlings treated with or without HSAL for 12 h. These genes should be presented the supplemental data. Furthermore, the expression of (DEGs) need to be confirmed by qRT-PCR, at least those iron transport-related genes presented in Fig 4d.
4. Supplementation of HSAL also appears rescue the growth defect of *Arabidopsis thaliana* caused by the deficiency of different nutrients, including Ca, Mg, Zn and Cu. The uptake of Ca and Mg by plant roots are thought to be in the ion form. So how does HSAL promote the plant growth, also in a chelation manner? I would suggest to discuss this point in the Discussion.
5. The statistical analysis were missing in Fig 3f and 3g, Fig 5d-g.

Reviewer #2 (Remarks to the Author):

The authors developed a hydrothermal reaction to modify lignin by increasing the ratio of phenolic hydroxyl groups to methoxy groups. The prepared lignin possessed the improved capacity to chelate metal ions. The results showed that the modification of HSAL can improve nutrient bioavailability and promote root length and plant growth. However, the starting lignin materials used in the study was just obtained from a standard Klason methods, an analytical protocol of biomass composition. This

method cannot be industrialized as it consumes more acids and is economically unviable. Additionally, there is no control experiment for lignin utilization. There are some other concerns that need to be addressed in the text.

1. The title should be changed or revised.
2. It is suggested to rewrite the abstract to present the key results and conclusions
3. Authors introduced more information of biorefinery and discussed the lignin potential from biorefinery industry in Introduction section. Since there are many lignin types prepared from different fractionations, why they selected "Sulfuric acid lignin (SAL)" as starting materials?
4. "Sulfuric acid lignin (SAL) represents insoluble lignin fragments and is a byproduct of the acid saccharification process on lignocellulosic materials for bio-fuel generation 20." This is wrong or confused statement. Acid pretreatment is not used for saccharification of carbohydrates in bio-fuel generation in biorefinery, and the lignin can not be obtained after acid pretreatment directly. Authors should provide the details of lignin preparation and why they name the lignin as sulfuric acid lignin? Actually, the methods used in the study to prepare lignin is just an composition analysis protocol and not an pretreatment approach employed in biorefinery.
5. The last paragraph of introduction should be rewritten
6. The language of the text need to be improved
7. "As expected, the HSAL powder can be completely dissolved in the neutral water (Fig.116 1a)." Did author mean the HSAL can dissolve into water at pH7.0 completely without any conditioning? Please provide the detailed information
8. Water solubility of lignin depends on the degree of depolymerization, de-methoxy, and phenolic hydroxyl content²². Did authors ignore carboxyl groups in lignin?
9. "these results suggest that thehydrothermal reaction not only breaks the carbon-oxygen-carbon bonds and carbon carbon bonds between lignin monomers but also most likely cleaves the alkyl-aryl ether bonds of the aromatic nucleus structure to reduce methoxy group and vacate the positions for phenolic hydroxyl groups, which leads to the water solubility of HSAL." Please provide the data of linkage degradation in this process
10. Did authors prepared SAL as a control for the evaluation of its chelation capability and other capabilities? Like in Figure 2
- 11.1. In Fig.2 d, the location of peaks 3360 and 3410 is seen to be reversed. Also, it would be better to label all the peaks mentioned in the main manuscript in Fig.2 e and d, so that the reader can compare them.
- 12.3. Line 136-240 Please further explain further the relationship between increasing the water solubility of lignin and the application of lignin in this paper. Also, what's the solubility of lignin in water before and after hydrothermal treatment?
13. Page 7 line 189, as lignin is a polydisperse polymer, the statement of "one molecular of HSAL can chelate more metals than EDTA." should be clarified.
14. It is suggested to illustrate the mechanism enhancing iron bioavailability by lignin.
15. In Figure 6, SAL is not an industrial lignin.

REVIEWER COMMENTS

Reviewer #1 (Remarks to the Author):

In this manuscript, Liu et al. nicely demonstrated that the water soluble hydrothermal sulfuric acid lignin (HSAL) can serve as an eco-friendly and economical nutrient chelator to improve plant nutrient bioavailability and enhance plant growth. This manuscript covers several aspects involved in material science, molecular science and plant nutrition, from the synthesis and characterization of HSAL to physiological roles of HSAL in promoting plant growth and the underlying molecular mechanism. I am not familiar with the part of the synthesis and characterization of HSAL but really enjoy in reading how they demonstrated HSAL functioning as an iron nutrient chelator to promote plant growth. By adding 0.05% HSAL, the authors nicely showed that HSAL did promote growth of several plant species. Taking the advantage of genome-wide transcriptomic analysis they found that HSAL mainly acts as a ferric iron chelator to increase iron bioavailability and thus promote plant growth under both iron limited or sufficient conditions. This conclusion was elegantly confirmed by using several Arabidopsis mutants, including the iron uptake transporter mutant *irt1*, the ferric chelate reductase mutant *fro2* and the phloem-specific iron transporter mutant *opt3*. Overall, this manuscript provided a sustainable route to utilize industrial lignin by conversion to HSAL which can potentially replace synthetic chelating agents and serve as a promising eco-friendly metal chelator to promote the profitability and sustainability of biorefineries and reduce the immission of CO₂ emissions. The experiments in the manuscript were well designed and the conclusion was convincing. I don't have any critical comments but a few suggestions as below.

Response: We appreciate the reviewer's encouraging comments.

Supplementation of 0.05% HSAL promotes the growth of both graminaceous plants (rice and maize) and nongraminaceous plant (*Arabidopsis thaliana*). These two type plants use different Fe acquisition strategies with Strategy I in nongraminaceous plants and Strategy II in graminaceous plants. It is reasonable that HSAL addition could promote the growth of *Arabidopsis*, the Strategy I plant, as HSAL-chelated Fe(III) can be reduced to Fe(II) and uptake by *IRT1*. This is also evidenced by using the *Arabidopsis irt1* and *fro2* mutants. For the Strategy II plants like rice, plants take up iron in the form of phytosiderophore chelated Fe(III), for example, deoxymugineic acid chelated Fe(III) for rice. So is it possible Strategy II plants could direct uptake the HSAL chelated Fe(III)? I would suggest to discuss this point in the Discussion.

Response: This is an interesting question. Thank you for this suggestion. It is suggested that the natural lignin-derived substances with high molecular size fraction ($M_w > 3500$) are not directly absorbed by plants (Nardi *et al.*, 2002). As the HSAL solution was collected carefully and dialyzed using a cellulose tube with a 3500 molecular weight cutoff (the average molecular weight is 7500), we don't think that Strategy II plants like rice could directly take up HSAL chelated Fe(III). Alternatively, ferric iron might be released from HSAL chelated complexes, and transported into root cells in the form of phytosiderophore chelated complexes.

As suggested by reviewer, we have added the discussion "In Strategy II plants like rice, roots take up iron in the form of phytosiderophore chelated ferric iron complexes. However, we do not think HSAL chelated ferric iron directly enters into plant cells, because the molecular weight of HSAL is higher than 3500 (we had collected HSAL using a cellulose tube with a 3500 molecular weight cutoff), and natural lignin-derived substances with a high molecular size fraction ($M_w > 3500$) has been suggested to be not directly absorbed by roots⁵¹. Alternatively, ferric iron might be released from HSAL chelated complexes, and transported into root cells in the form of phytosiderophore chelated complexes." (Lines 480-488, page15).

References:

Nardi, S., *et al.* Physiological effects of humic substances on higher plants. *Soil Biol. Biochem.* **34**, 1527-1536 (2002).

In Fig 6, rice, a Strategy II plant, was used on the top right of the schematic diagram but the Strategy I iron acquisition pathway was showed in the bottom right. This inconsistency must be corrected.

Response: We thank the reviewer for pointing out this issue. We have corrected it in the revised Fig. 6.

Numerous differentially expression genes (DEGs) were identified in the root tip of 7-day-old seedlings treated with or without HSAL for 12 h. These genes should be presented the supplemental data. Furthermore, the expression of (DEGs) need to be confirmed by qRT-PCR, at least those iron transport-related genes presented in Fig 4d.

Response: Thank you for the suggestion. We have added the list of DEGs as Supplementary Dataset 1. Furthermore, the expression of these iron transport-related genes presented in Fig 4d were confirmed with qRT-PCR analysis. The result of qRT-PCR showed the pattern of these iron transport-related genes is consistent with that of RNA-seq analysis. The experiments were repeated independently and showed the same pattern. This result was shown as Supplementary Fig. 6d.

Supplementary Fig. 5d | HSAL additive increases expression of iron transport-related genes in rice root tips. After grown in hydroponics for 7 days, rice seedlings were treated with and without 0.05% HSAL for 12 h. Root tips were harvested to extract RNA for qRT-PCR. The graphs depict the mean values with standard deviation (error bars). Individual data points are plotted. p-value is computed using Student's t-test (n=6) on comparing control and 0.05% HSAL treatments.

Supplementation of HSAL also appears rescue the growth defect of *Arabidopsis thaliana* caused by the deficiency of different nutrients, including Ca, Mg, Zn and Cu. The uptake of Ca and Mg by plant roots are thought to be in the ion form. So how does HSAL promote the plant growth, also in a chelation manner? I would suggest to discuss this point in the Discussion.

Response: We thank the reviewer for this thoughtful comment. Application of HSAL rescued the growth defect of *Arabidopsis thaliana* caused by the deficiency of different nutrients, including Fe, Ca, Mg, Zn, and Cu (Supplementary Fig. 8), and also increased the total amount of these metal nutrients especially for Fe and Ca (Supplementary Fig. 9). These results indicate that this function of HSAL is not specific to iron, which is consistent the metal chelation capacity of HSAL shown in Fig 2. As EDTA-based polymetallic element chelators have been widely developed to improve the bioavailability of metal nutrients through increasing the mobility of metal ions. Therefore, we think HSAL is most likely to improve the mobility of these ions in the growth medium by increasing metal nutrient diffusion to the root surface and/or improving the mobility of these ions in root apoplastic space through its chelation capacity, which leads to increase bioavailability of these metal nutrients for plants.

As suggested by the reviewer, we have added this text to the discussion “Application of HSAL also rescued the growth defect of *Arabidopsis thaliana* caused by the deficiency of different nutrients, including Ca, Mg, Zn, and Cu (Supplementary Fig. 8), and also increased the total amount of these metal nutrients especially Ca in plant tissues (Supplementary Fig. 9). These results therefore indicate that the function of HSAL is not specific to iron, which is consistent with the metal chelation capacity of HSAL. We think HSAL most likely improves the mobility of these ions in the growth medium to increase metal nutrient diffusion to the root surface and/or to avoid iron precipitating in cell walls through its chelation capacity, which leads to increased bioavailability of these metal nutrients for plants.” in **lines 489-511, page15-16**.

The statistical analysis was missing in Fig 3f and 3g, Fig 5d-g.

Response: We thank the reviewer for the suggestion. As suggested by the reviewer, we have added the statistical analysis of Fig 3f and 3g, Fig 5d-g.

Reviewer #2 (Remarks to the Author):

The authors developed a hydrothermal reaction to modify lignin by increasing the ratio of phenolic hydroxyl groups to methoxy groups. The prepared lignin possessed the improved capacity to chelate metal ions. The results showed that the modification of HSAL can improve nutrient bioavailability and promote root length and plant growth. However, the starting lignin materials used in the study was just obtained from a standard Klason methods, an analytical protocol of biomass composition. This method cannot be industrialized as it consumes more acids and is economically unviable. Additionally, there is no control experiment for lignin utilization. There are some other concerns that need to be addressed in the text.

Response: We thank the reviewer for pointing out these issues in the current manuscript and give us an opportunity to clarify these problems.

However, the starting lignin materials used in the study was just obtained from a standard Klason methods, an analytical protocol of biomass composition. This method cannot be industrialized as it consumes more acids and is economically unviable.

Response: We thank the reviewer for the correction. The starting lignin material SAL is indeed not directly from industrial lignin, but obtained from a standard Klason method which was used to simulates the lignin byproduct from lignocellulosic biorefinery by using concentrated sulfuric acid (Megawati et al., 2015). Therefore, in the revised manuscript, we have clarified the statement in the Method section to make it clear as followed: “To obtain high-purity sulfuric acid lignin (SAL) that simulates the lignin byproduct from lignocellulosic biorefinery by using concentrated sulfuric acid, we used the Klason method to fully remove cellulose and other carbohydrates in wood powder by using concentrated sulfuric acid hydrolysis.” (Lines 550-553, page 16). We have also replaced “industrial lignin” with “Sulfuric acid lignin” in Figure 6.

References:

Megawati, et al., Sulfuric acid hydrolysis of various lignocellulosic materials and its mixture in ethanol production. Biofuels, 6, 331 - 340 (2015).

Additionally, there is no control experiment for lignin utilization.

Response: Since SAL is completely insoluble in water at room temperature and can't be a good control for plant growth, we directly used water as control for HSAL utilization for plant growth. In order to exclude the possibility of HSAL being a direct nutrient resource (such as iron) rather than its chelation function for plant growth, we measured the metal nutrient content in 0.05% HSAL. The result showed that the metal nutrient content in 0.05% HSAL can be ignored for plant growth compared to the nutrient content in growth medium and nutrient accumulation in plants (Supplementary Table 2-4).

The title should be changed or revised.

Response: The title has been changed into “A lignin-derived material improves plant nutrient bioavailability and growth through its metal chelate capacity”

It is suggested to rewrite the abstract to present the key results and conclusions

Response: As suggested by the reviewer, we have worked the key results and conclusions in the abstract of the revised manuscript.

Authors introduced more information of biorefinery and discussed the lignin potential from biorefinery industry in Introduction section. Since there are many lignin types prepared from different fractionations, why they selected “Sulfuric acid lignin (SAL)” as starting materials?

Response: Thank you for the suggestion. In the revised manuscript, we have added more information in Introduction section (**Lines 92-102, page 4**) to show why we selected Sulfuric acid lignin (SAL) as a starting material:

“Sulfuric acid hydrolysis and enzymatic hydrolysis are two major strategies to produce monosaccharides in commercial biorefinery²⁰. Sulfuric acid lignin (SAL, generated from sulfuric acid hydrolysis) and enzymatic hydrolysis lignin (generated from enzyme-mediated hydrolysis), are the major lignin byproducts from biorefinery²¹. Both types of lignin show very low solubility

in both water and organic solvents due to their high molecular weights and highly condensed structures²² (this is stronger in SAL due to the repolymerization and condensation reactions). This limits the valorization of the waste lignin and restricts the sustainable development of lignocellulosic biorefinery²³⁻²⁴. Therefore, SAL represents an acid-insoluble lignin residual portion obtained from polysaccharide conversion of lignocellulosic biomass by conducting acidolysis using concentrated sulfuric acid²⁵.”

“Sulfuric acid lignin (SAL) represents insoluble lignin fragments and is a byproduct of the acid saccharification process on lignocellulosic materials for bio-fuel generation 20.” This is wrong or confused statement. Acid pretreatment is not used for saccharification of carbohydrates in bio-fuel generation in biorefinery, and the lignin can not be obtained after acid pretreatment directly. Authors should provide the details of lignin preparation and why they name the lignin as sulfuric acid lignin? Actually, the methods used in the study to prepare lignin is just a composition analysis protocol and not an pretreatment approach employed in biorefinery.

Response: We thank the reviewer for pointing this out. We called the starting lignin sulfuric acid lignin (SAL), because this lignin is usually generated from lignocellulosic biorefinery by using sulfuric acid hydrolysis. In our study, SAL is the lignin produced by sulfuric acid hydrolysis rather than by acid pretreatment of lignocellulosic biorefinery. In order to avoid confusion with this statement, we added more information about SAL in the Introduction section (**Lines 100-102, page 4**). Additionally, we have revised the sentence as ‘SAL represents an acid-insoluble lignin residual portion obtained from polysaccharide conversion of lignocellulosic biomass by conducting acidolysis using concentrated sulfuric acid²⁵.’ As suggested by the reviewer, we have also added more information of lignin preparation in the Method section of the revised manuscript as follow: “In order to obtain high-purity sulfuric acid lignin (SAL) that simulates the lignin byproduct from lignocellulosic biorefinery by using concentrated sulfuric acid, we used the Klason method to fully remove cellulose and other carbohydrates in wood powder by using concentrated sulfuric acid hydrolysis.” (**Lines 550-553, page 16**).

The last paragraph of introduction should be rewritten

Response: In the revised manuscript, we have added more information of sulfuric acid lignin at the beginning of the last paragraph of Introduction section (**Lines 92-102, page 4**) as followed:

“Sulfuric acid hydrolysis and enzymatic hydrolysis are two major strategies to produce monosaccharides in commercial biorefinery²⁰. Sulfuric acid lignin (SAL, generated from sulfuric acid hydrolysis) and enzymatic hydrolysis lignin (generated from enzyme-mediated hydrolysis), are the major lignin byproducts from biorefinery²¹. Both types of lignin show very low solubility in both water and organic solvents due to their high molecular weights and highly condensed structures²² (this is stronger in SAL due to the repolymerization and condensation reactions). This limits the valorization of the waste lignin and restricts the sustainable development of lignocellulosic biorefinery²³⁻²⁴. Therefore, SAL represents an acid-insoluble lignin residual portion obtained from polysaccharide conversion of lignocellulosic biomass by conducting acidolysis using concentrated sulfuric acid²⁵.”

The language of the text needs to be improved

Response: Thank you for the suggestion. We have carefully edited the language throughout the revised manuscript. These edits are shown with track changes.

“As expected, the HSAL powder can be completely dissolved in the neutral water (Fig. 1a).” Did author mean the HSAL can dissolve into water at pH7.0 completely without any conditioning? Please provide the detailed information

Response: Thank you for the suggestion. HSAL can be dissolved completely at room temperature in neutral water at pH7.0. We have changed this sentence into “As expected, the HSAL powder can be completely dissolved in neutral water at room temperature (The solubility is about 3 grams per 100 ml)” in the revised manuscript (**Lines 131-132, page 5**).

Water solubility of lignin depends on the degree of depolymerization, de-methoxy, and phenolic hydroxyl content²². Did authors ignore carboxyl groups in lignin?

Response: We thank the reviewer for pointing this issue out. We have changed this sentence into “Water solubility of lignin depends on the degree of depolymerization, de-methoxy, phenolic hydroxyl content and carboxyl groups²⁷. Since the carboxyl groups will undergo an intense and thorough decarboxylation reaction under this hydrothermal reaction²⁸, the carboxyl groups can be ignored in HSAL²²” in the revised manuscript (**Lines 133-136, page 5**)

“these results suggest that the hydrothermal reaction not only breaks the carbon-oxygen-carbon bonds and carbon carbon bonds between lignin monomers but also most likely cleaves the alkyl-aryl ether bonds of the aromatic nucleus structure to reduce methoxy group and vacate the positions for phenolic hydroxyl groups, which leads to the water solubility of HSAL.” Please provide the data of linkage degradation in this process

Response: We thank the reviewer to raise this issue. We did not fully introduce the background of HSAL generated by this this hydrothermal reaction. In our previous study, we have shown this hydrothermal reaction can cleave the carbon-oxygen-carbon bonds, carbon-carbon bonds and alkyl-aryl ether bonds of the aromatic nucleus structure by using oligomeric lignin model compounds with β -O-4 linkage and polymeric lignin model compounds (Matsushita et al., 2013; Liu et al., 2018). This finding was further supported by the great change of the average molecular weight, the methoxyl group, the phenolic hydroxyl content and the alcohol hydroxyl content of HSAL compared to SAL (Figure 1).

In the revised manuscript, we have added “In previous studies, we have shown that this hydrothermal reaction can cleave carbon-oxygen-carbon bonds, carbon-carbon bonds and alkyl-aryl ether bonds of the aromatic nucleus structure by using oligomeric lignin model compounds with β -O-4 linkage²² and polymeric lignin model compounds²⁹. This finding was partly supported by the reduction of the average molecular weight from 57KDa of SAL to 7.5KDa of HSAL.” (**Lines 136 to 141, page 5**). We also modified this sentence into “Taken together, these results supported the notion that the hydrothermal reaction not only breaks the carbon-oxygen-carbon bonds and carbon-carbon bonds between lignin monomers but also most likely cleaves the alkyl-aryl ether bonds of the aromatic nucleus structure to reduce the methoxy group and to vacate the positions for phenolic hydroxyl groups, which leads to the water solubility of HSAL.” (**Lines 166-170, page 6**)

References:

Matsushita, Y., et al. Hydrothermal reaction of sulfuric acid lignin generated as a by-product during bioethanol production using lignocellulosic materials to convert bioactive agents. Ind. Crops Prod. 42, 181-188 (2013).

Liu, Q., et al. Effects of hydrothermal reaction of sulfuric acid lignin from Cryptomeria japonica for industrial utilization. BioRes. 13, 7805-7825 (2018).

Did authors prepare SAL as a control for the evaluation of its chelation capability and other capabilities? Like in Figure 2

Response: We also would have liked to prepare SAL as a control for all experiments for HSAL used in our study. However, due to its highly condensed chemical structure, SAL is completely insoluble in water at room temperature (Figure 1a, Matsushita et al., 2013). The water insolubility limits SAL to be used as plant nutrient chelator or regulator. Therefore, we tried to modify the structure of SAL to increase its water solubility using a hydrothermal treatment.

References:

Matsushita, Y., et al. Hydrothermal reaction of sulfuric acid lignin generated as a by-product during bioethanol production using lignocellulosic materials to convert bioactive agents. Ind. Crops Prod. 42, 181-188 (2013).

In Fig.2 d, the location of peaks 3360 and 3410 is seen to be reversed. Also, it would be better to label all the peaks mentioned in the main manuscript in Fig.2 e and d, so that the reader can compare them.

Response: Thank you for the correction and suggestion. In the revised manuscript, we have reversed the location of peaks 3360 and 3410 in Fig. 2d. All the peaks mentioned in the main manuscript in Fig.2 e and d have been labeled.

Line 136-240 Please further explain further the relationship between increasing the water solubility of lignin and the application of lignin in this paper. Also, what's the solubility of lignin in water before and after hydrothermal treatment?

Response: We thank the reviewer for providing an opportunity to further explain the relationship between the water solubility of lignin and its regulation role in metal nutrient bioavailability and plant growth. Take iron as an example: iron is very enriched in soils, but iron is almost insoluble

and immobile due to its oxidized and precipitated state in alkaline soils. This leads to low bioavailability of iron for plant uptake, and causes iron deficiency problem in plants in alkaline soils. Application of metal chelators such as EDTA is a common approach to improve iron bioavailability in agriculture, but the synthetic EDTA is not bio-degradable. Lignin-derived materials have the potential to be used as bio-degradable metal chelators. However, most of industrial lignin is not water soluble, and cannot be directly used as a regulator for plant growth. Sulfuric acid lignin (SAL) is a structurally highly condensed lignin and insoluble in water. After hydrothermal treatment of SAL, HSAL can be completely dissolved at room temperature in neutral water at pH7.0 (Solubility is about 3 grams per 100 ml) (Figure 1a, Matsushita et al., 2013).

References:

Matsushita, Y., et al. Hydrothermal reaction of sulfuric acid lignin generated as a by-product during bioethanol production using lignocellulosic materials to convert bioactive agents. Ind. Crops Prod. 42, 181-188 (2013).

Page 7 line 189, as lignin is a polydisperse polymer, the statement of “one molecular of HSAL can chelate more metals than EDTA.” should be clarified.

Response: Thank the reviewer for the correction. We have revised “one molecular of HSAL can chelate more metals than EDTA” into “equimolar amounts of HSAL can chelate more metals than EDTA”. (Lines 231-232, page 8)

It is suggested to illustrate the mechanism enhancing iron bioavailability by lignin.

Response: Thank you for the suggestion. The illustration for the mechanisms enhancing bioavailability by HSAL has been revised in Figure 6. We also added a new paragraph in Discussion section to discuss “HSAL enhances plant nutrient bioavailability through its metal chelate capacity” in lines 471 – 511, page 14-15.

In Figure 6, SAL is not an industrial lignin.

Response: Thank you for pointing this issue. We have replaced “industrial lignin” with “Sulfuric acid lignin” in Figure 6.

Reviewer 3

The manuscript by Liu et al. is an interesting work dealing with the use of a lignin derived material (HSAL) that seems to improve iron uptake by plants. HSAL derives from lignin after a hydrolytic process based on sulfuric acid, followed by a hydrothermal treatment conducted under basic conditions. The characterization of the product indicates a lower degree of polymerization with respect to the starting lignin, as well as an increase of the hydroxyl functionalities. HSAL has a beneficial effect on the growth of different types of plants. The authors attribute such an effect to the chelation property of HSAL towards iron. As evidence of the chelation properties of HSAL, the authors have prepared Fe-containing and Ca-containing materials starting from HSAL, whose characterization based on FTIR, XPS and complexometric titration, is considered to offer evidence of such a behavior.

In my opinion, the manuscript is well written and well organized, and the topic covered is of high interest for all the researchers involved in the development of new and sustainable agricultural materials/practices, which represents a hot topic in the modern research. For this reason, the manuscript fits perfectly the scope of the journal.

However, it is opinion of this reviewer that before the manuscript can become suitable for publication, the following concerns must be clearly addressed.

Response: We appreciate the reviewer's thoughtful comments on our work and for emphasizing the novelty and importance of our findings.

1) My main criticism is on characterization of the metal-containing materials. It is mandatory that the authors exclude the possibility that the invoked "metal complexes" are not hybrid materials composed by HSAL and inorganic phases, such as insoluble metal oxides, maybe present as micro or nano-materials embedded in the polymeric matrix. These inorganic phases could form because of the presence of free NaOH or because of the basicity of HSAL itself once dissolved in water. The FTIR characterization cannot be definitive in discharging such a possibility and, moreover, the XPS signals recorded by the authors correspond to those of pure Fe₂O₃ and CaO. Important indications could derive by interrogation of the isolated metal-containing materials by XRPD

analysis or, even better, to TEM-ED analysis. In this regard, the following papers could be informative:

- ACS Sustainable Chemistry&Engineering, 2019, 7, 3213, doi: 10.1021/acssuschemeng.8b05135

- Advanced Sustainable Systems, 2022, 2200108, doi: 10.1002/adsu.202200108

Response: Thank you very much for pointing out the possibility that the metal-containing materials of HSAL chelated metals are hybrid materials composed by HSAL and inorganic phases. As the reviewer mentioned, these inorganic phases might form hybrid materials due to the presence of free NaOH or because of the basicity of HSAL itself once dissolved in water.

As suggested by the reviewer, we took HSAL-FeCl₃ as example to perform XRPD and TEM-EDS analysis (Sinisi et al., 2018, Gazzurelli et al., 2022) to examine whether the HSAL-metal chelation complexes form hybrid materials composed by HSAL and inorganic phases in the revised manuscript. As shown in Supplementary Fig. 2, the XRPD trace of HSAL and HSAL-FeCl₃ showed a smooth crest. The TEM-EDS image showed no bright spot aggregation distribution in HSAL and HSAL-FeCl₃ complexes. Taken together, these results exclude the possibility that chelated metal compounds form oxide crystals embedded in the polymeric matrix of HSAL. These results have also been described in the text of the revised manuscript (**Lines 234-245, page 8**).

Supplementary Fig. 2 | XRPD and TEM-EDS analysis of HSAL and HSAL-FeCl₃ complexes.

(a) Overlap of XRPD plots of HSAL powder and HSAL-FeCl₃ complexes. (b) HAADF-STEM and TEM-EDS analysis and carbon distribution on the surface of HSAL powder. (c) HAADF-STEM and TEM-EDS analysis and carbon-iron distribution on the surface of HSAL-FeCl₃ powder.

References:

Sinisi, V., et al. A Green Approach to Copper-Containing Pesticides: Antimicrobial and Antifungal Activity of Brochantite Supported on Lignin for the Development of Biobased Plant Protection Products. *ACS Sustainable Chem. Eng.* **7**, 3213-3221 (2018).

Gazzurelli, C., et al. Exploiting the Reducing Properties of Lignin for the Development of an Effective Lignin@Cu₂O Pesticide. *Advanced Sustainable Systems*, **6**, 2200108. (2022).

The exact amounts of Fe and Ca used to prepare the corresponding materials must be reported. The quantification of the metal content in the Fe and Ca-containing materials is required, for instance based on ICP analysis, in order to quantify the metal uptake.

Response: We thank the reviewer for the suggestion. We have added the amounts of FeSO₄ (15.19g), FeCl₃(16.22g), or CaCl₂ (11.09g) in the revise method (**Lines 615-617, page 19**) as follow: “To obtain HSAL-metal chelation complexes, 0.2 g HSAL was dissolved in 20 ml of

ultrapure water, and 0.1M of FeSO₄ (15.19g), FeCl₃ (16.22g), or CaCl₂ (11.09g) respectively were added.”. As suggested by the reviewer, we also performed elemental analysis of HSAL and HSAL-metal complexes. The results showed that iron content is clearly higher in HSAL-FeSO₄ and HSAL-FeCl₃, while calcium content is also much higher in HSAL-CaCl₂ compared to HSAL (Supplementary Table 1). This result supported that HSAL has successfully chelated with iron and calcium. This information has been added in the revised manuscript (**Lines 182-186, page 7**).

Supplementary Table 1. The Ca and Fe content in HSAL-metal complexes.

	Ca ($\mu\text{g. g}^{-1}$)	Fe ($\mu\text{g. g}^{-1}$)
HSAL	70.86 \pm 4.57	12.13 \pm 2.55
HSAL-FeSO ₄	341.8 \pm 88.11	6711 \pm 640.0
HSAL-FeCl ₃	870.8 \pm 93.50	2443 \pm 289.7
HSAL-CaCl ₂	10047 \pm 974.7	54.27 \pm 3.20

For complexometric titration the authors used metal salts different from those used to prepare the metal-containing materials. Please, explain why.

Response: We thank the reviewer for the thoughtful comments. FeSO₄ and FeCl₃ are two representative iron salts used in plant growth medium, so they were chosen to prepare the HSAL-metal chelated complexes. In the complexometric titration reaction accompanied by acidification, the formed metal complexes change with pH, resulting in instability of metal complexes. NH₄Fe(SO₄)₂ is usually selected for complex titration to ensure the stability of complex titration. That is why different metal salts were chosen to complexometric titration and preparation of the metal-containing materials.

It is opinion of this reviewer that these criticisms are not against the interpretation that HSAL has chelating properties toward iron, but rather that the synthesized materials cannot be used to demonstrate the chelating properties of HSAL without first excluding the formation of low soluble inorganic phases.

Response: Thank you for the constructive comment. As suggested by the reviewer, we have performed XRPD and TEM-EDS analysis of HSAL and HSAL-FeCl₃ and showed that both of them did not have obvious crystallization and iron ion was evenly distributed on the surface of the

carbon baseplate (Supplementary Fig. 2). This result supported that HSAL did not form hybrid materials composed by HSAL and inorganic phases.

Other minor points:

1) page 7, line 165: p must be lowercase

Response: Thank you for the correction. It has been included in the revised manuscript.

2) page 7, line 180: I find hard that the IR stretching band of Ca-Cl bond falls at 3640 cm⁻¹

Response: Thank you for the correction. We compared FT-IR spectra of CaCl₂ and HSAL chelated with CaCl₂. The FT-IR stretching band of Ca-Cl bond falls at 3424 cm⁻¹ for the Ca-Cl bond of CaCl₂ was shifted to 3400 cm⁻¹ for the chelated complexes of HSAL with CaCl₂ (Fig. 2e). The sharp stretching band falls at 3640 cm⁻¹ likely corresponds to the O-H bonds from calcium hydroxide (Liang et al., 2018) due to calcium chloride combining remaining hydroxide in the chelated complexes.

The description has been revised as “Moreover, in the chelated complex of HSAL with CaCl₂, two new bands in the FT-IR spectrum were observed around 1200 cm⁻¹ and 3640 cm⁻¹, which can be likely assigned to the C-O of the guaiacol stretching and the O-H bonds from calcium hydroxide³³ due to calcium chloride combining remaining hydroxide in the chelated complexes, respectively.”

References:

Liang, Z., et al. Ion-triggered calcium hydroxide microcapsules for enhanced corrosion resistance of steel bars. RSC Adv. 8, 39536 - 39544. (2018).

3) page 10, lines 277 and 280: HASL must be replaced with HSAL

Response: We thank the reviewer for pointing this out. We have corrected ‘HASL’ with ‘HSAL’ throughout the manuscript.

4) page 16, line 446: ^{1H}NMR and not 1HNMR

Response: ‘¹H NMR’ has been corrected to ‘¹H NMR’

5) Figure 1g: remove the subscript 3 associated with the OH group in the structure corresponding to “Softwood lignin”

Response: Thanks for pointing it out. We have removed the subscript 3 associated with the OH group in the structure corresponding to “Softwood lignin” in Figure 1g.

REVIEWERS' COMMENTS

Reviewer #1 (Remarks to the Author):

The authors have performed additional analysis and revised the manuscript according to my suggestions that I raised in the last review process. All of my concerns have addressed so I have no more comments. This manuscript have been well improved in current version particularly with the newly collected data according to other reviewers' suggestions.

Reviewer #2 (Remarks to the Author):

Authors carefully revised the manuscript to response the comments from Reviewers. The manuscript should be considered for publication.

Reviewer #3 (Remarks to the Author):

It is opinion of this reviewer that the data added (XRPD, TEM and ICP analysis) and changes made by the authors to the previous submission are sufficient to address my concerns. I now find the manuscript fully acceptable.

REVIEWERS' COMMENTS

Reviewer #1 (Remarks to the Author):

The authors have performed additional analysis and revised the manuscript according to my suggestions that I raised in the last review process. All of my concerns have addressed so I have no more comments. This manuscript has been well improved in current version particularly with the newly collected data according to other reviewers' suggestions.

Response: We appreciate the reviewer's positive comments on our manuscript.

Reviewer #2 (Remarks to the Author):

Authors carefully revised the manuscript to response the comments from Reviewers. The manuscript should be considered for publication.

Response: We appreciate the reviewer's positive comments on our manuscript.

Reviewer #3 (Remarks to the Author):

It is opinion of this reviewer that the data added (XRPD, TEM and ICP analysis) and changes made by the authors to the previous submission are sufficient to address my concerns. I now find the manuscript fully acceptable.

Response: We appreciate the reviewer's positive comments on our manuscript.